# Pathways to scale up early childhood programs: A scoping review of Reach Up and Care for Child Development

Gabriela Buccini[1]*, Lily Kofke[2☯], Haley Case[2☯], Marina Katague[2‡], Maria Fernanda Pacheco[3‡], Rafael Pérez-Escamilla[2]

1 Department of Social and Behavioral Health, University of Nevada Las Vegas School of Public Health, Las Vegas, Nevada, United States of America, 2 Department of Social and Behavioral Sciences, Yale School of Public Health, New Haven, Connecticut, United States of America, 3 Yale College, New Haven, Connecticut, United States of America

☯ These authors contributed equally to this work.
‡ MK and MFP also contributed equally to this work.
* gabriela.buccini@unlv.edu

**Data Availability Statement:** All data generated and analyzed during this study are included in the

## Abstract

Evidence-based early childhood development (ECD) programs that strengthen nurturing parenting skills and promote early stimulation, such as Reach Up (RU) and Care for Child Development (CCD), are critical investments for interrupting cycles of intergenerational poverty; however, the implementation impact of these programs varies greatly globally. Analyzing systematically the evidence on the implementation pathways based on contexts (i.e., external and internal influences on intervention implementation), implementation strategies (i.e., mechanisms used to promote program initiation, design, and delivery with existing systems), and implementation outcomes (i.e., related to the implementation goals) can increase the likelihood of implementation success. Our scoping review aimed to identify implementation pathways of RU and CCD programs in low- and middle-income countries. A search in English, Spanish, and Portuguese of grey literature and five databases of peer reviewed literature; from inception through July 16, 2022, yielded 2,267 publications. Using predetermined eligibility criteria, 75 records yielded implementation details for 33 programs across 23 low- and middle-income countries. Two reviewers independently extracted program data on context, implementation strategies, and implementation outcomes following a program theory. A thematic analysis identified 37 implementation strategies across six "building blocks of implementation": program emergence, intersectoriality, intervention characteristics, workforce, training, and monitoring systems. Implementation pathways across building blocks are highly influenced by contextual factors, such as infrastructure, social norms, and the target population's demand and interest, which may shape different implementation outcomes. Six 'building blocks' shaping implementation pathways of CCD and RU in LMICs were identified. The careful consideration of context and use of intentional evidence-based planning can enable the successful implementation of ECD nurturing care interventions. We recommend the use of the ECD Implementation Checklist for Enabling Program Scale Up to guide decision-

published article and its supplementary information files.

**Funding:** Research reported in this publication was supported by the Eunice Kennedy Shriver National Institute for Child Health and Development (NICHD) of the National Institutes of Health under award number R00HD097301 (PI: Buccini). The PI received salary support for conducting the research reported in this publication. The funder had no role in the design of the study and collection, analysis, and interpretation of data. The content is solely the responsibility of the authors and does not necessarily represent the official views of the National Institutes of Health.

**Competing interests:** The authors declare that they have no competing interests.

making regarding context and implementation strategies to support implementation outcomes and subsequent ECD program success.

## Introduction

Investing in early childhood development (ECD) is a global priority to meet the 2030 Sustainable Development Goals [1–4]. In low- and middle-income countries (LMICs), 43% of children under the age of five are at risk of suboptimal development due to accumulated adverse experiences including poverty, food insecurity, neglect, and violence [4, 5]. A nurturing care approach can prevent or mitigate the negative consequences of such experiences by cultivating a safe, secure, and stimulating environment in which children have opportunities to learn and interact with caregivers who are emotionally supportive, sensitive, and responsive to their needs [4–6]. Therefore, investments in ECD nurturing care programs are essential for breaking the intergenerational cycles of poverty and ultimately reducing inequities since gestation, and during infancy and early childhood [7–9].

The Reach Up and Learn Early Childhood Parenting Program (RU) and Care for Child Development (CCD) are examples of evidence based ECD nurturing care programs extensively implemented in LMICs through integration into existing services across sectors such as health, nutrition, education, and child protection. Both programs guide or coach caregivers on how to observe and engage in interactive and stimulating activities with their children to promote their motor, cognitive-language, and social-emotional skills [10, 11]. RU is based on the stimulation arm of the Jamaica Home Visit intervention and is backed up by substantial evidence documenting its positive impact on education, mental health, income, and reductions in aggressive behavior [12–15]. CCD was developed by the World Health Organization (WHO) and UNICEF, and it is based on behavior change counseling methods to h empower caregivers to build stronger relationships with their young children through nurturing care [10, 16]. Caregivers may receive counseling during home visits or clinic consultations, or during parent, nutrition education, or other group sessions [10, 16]. Together, both programs have been implemented in over 20 countries [11, 16]. Details on the RU and CCD programs are described following the Template for Intervention Description and Replication (TIDieR) [17] (S1 Table).

Although RU and CCD's effectiveness has been well documented, efforts to scale up such programs have varied greatly in their aims, scope, and impact globally [18–21]. A recent global review on CCD points out the need for additional research to fill in gaps in knowledge on how to improve program effectiveness and process outcomes such as implementation fidelity and quality, and acceptance [22], which is particularly important for guiding scaling up in the context of the complex adaptative systems that need to be navigated. Scaling up is defined as expanding coverage and quality of a specific service to larger populations or broader geographical areas aiming at 'maximizing the reach and effectiveness of an intervention, leading to a sustained impact on outcomes [23]. Successful scale up requires evidence for replicability and adaptations to local contexts, to maximize effectiveness, sustainability, and long-term impact As expected given how complex this is in the real world, several LMICs have faced challenges to implementing RU and CCD methodologies into large-scale multisectoral nurturing care interventions [24–27].

Systematizing the analysis of evidence regarding the implementation pathways between contexts (i.e., external and internal influences on intervention implementation), implementation strategies (i.e., mechanisms used to promote program initiation, design, and delivery with

existing systems), and implementation outcomes (i.e., related to the program goals) can increase the likelihood of scale up success [28]. Further such analysis of the evidence, can inform the design and development of context-sensitive integrated interventions that are critical for scaling up interventions in an effective and sustainable way, especially in resources-constrained settings such as LMICs [18, 20, 21]. Yet, as far as we know there have been no previous attempts to systematically identify the implementation pathways and key approaches needed to successfully scale up ECD programs such as RU and CCD across diverse settings.

The specific aim of this study was to conduct a scoping review to systematically identify the context, strategies, and outcomes shaping implementation pathways of RU and CCD nurturing care programs targeting children under 5 years old in LMICs. Based on our review, we highlight implementation pathways theories for scaling up ECD nurturing care programs and propose an implementation checklist for enabling programming for nurturing care globally. To our knowledge this is the first analysis of its kind greatly expanding the knowledge in this area [18–22].

## Materials and methods

We conducted a scoping review, because the present study was designed to identify a heterogeneous body of literature regarding a concept that is very broad in scope [29]. As recommended, we present findings following the Preferred Reporting Items for Systematic reviews and Meta-Analyses extension for Scoping Reviews (PRISMA-ScR) guidelines (S1 Checklist). This review was preregistered in Prospero (CRD42020199294).

### Information sources

The following databases were searched: PubMed, Web of Science, CINAHL, Ovid Global Health, and LILACS. The following grey literature sources were also searched: WHO, World Bank, UNICEF, Spring Impact, Reach Up and Learn Website, Grand Challenges, Nurturing Care Framework website, Bernard Van Leer Foundation, Results for Development, and Fundação Maria Cecília Souto Vidigal. There were no restrictions regarding language or publication period. Electronic database searches were supplemented by cross-checking the reference list of included articles and systematic reviews identified during the title and abstract screening. Searches were conducted on July 26, 2020, and updated on July 16, 2022.

### Search strategy

The search strategy was developed following the Population, Concept, and Context (PCC) strategy [30]. In this scoping review, the **population** was CCD and RU programs targeting children under five years old, the **concept** consisted of nurturing care integrated programs and the **context** encompassed implementation in LMICs. Medical Subject Headings (MeSH) terms and Health Science Descriptors (DeCS) were selected to operationalize the PCC search strategy. Search terms were designed in English and translated into Portuguese and Spanish. The search strategy was designed for the PubMed database and adapted for the other databases. For each website, the search strategy was adapted according to its available resources and search interface. Search strategies were validated by a librarian with expertise in Public Health and reviews (see S2 Table).

### Eligibility criteria

We included human subjects' studies related to RU or CCD programs targeting children under age five and their caregivers in LMICs. All study designs, peer-reviewed, and non-peer-

reviewed articles, reports, program manuals, and websites that presented information related to RU or CCD in LMICs were included. There was no restriction regarding language.

We excluded records on programs that were not based on RU or CCD, were not conducted in LMICs, or did not have implementation outcomes details. Details on implementation outcomes were deemed sufficient when they included program-specific information on adaptation or scaling [31–33] or when they detailed implementation outcomes as distinct from service systems and clinical treatment outcomes [34]. Study protocols were excluded if trial and/or implementation results were not available.

## Selection of sources of evidence

After the removal of duplicates, titles and abstracts were dual screened for the inclusion criteria by four reviewers (HC, LK, MK, and MFP) previously standardized against one another. The full texts of all potentially relevant citations were retrieved and independently assessed for eligibility using the predefined inclusion and exclusion criteria. Studies determined to be potentially relevant or whose eligibility was uncertain were retrieved for full-text review. Any disagreements were solved through a consensus process and, if necessary, consulting a senior reviewer (GB).

## Data extraction

Country program data components across contexts, strategies, and outcomes were extracted by four reviewers (HC, LK, MK, and MFP). Data extraction items followed an initial program theory [1, 4, 5, 18, 35–39] (S1 Fig) that was then revised following the literature review phase. Data extraction was piloted by having reviewers extract from the same record and then compare their extraction results with one another. Disagreement was resolved by discussion, and where agreement could not be reached, a senior reviewer (GB) was consulted. The extraction process was determined to be sufficient based on reviewers' consensus following the first two record extractions. Based on pilot testing, the data extraction sheet was improved and then used throughout the qualitative data collection. Data items were organized across context (10 items), implementation strategies (23 items), and implementation outcomes (11 items), and then were used by a single reviewer to extract data from each country program record (S3 Table). The definition of implementation outcomes was adapted from Proctor et al. [34] and can be found in Table 1. Qualitative data on implementation strategies and implementation outcomes were organized by implementing sites within each country (see S4–S7 Tables for the qualitative summary).

## Data synthesis

We adapted a thematic synthesis method used for scoping and systematic reviews [40–42] to identify inter-relationships between context, strategies, and outcomes characterizing the pathways to implementation across implementing sites. The study involved two stages of analysis: (1) coding line-by-line data to develop themes across context, strategies, and outcomes (i.e., conceptual codes) similar to those reported in primary studies, and (2) the generation of relationship codes to develop hypothesized implementation pathways theory hereafter "building blocks of implementation" within studies but apparent between studies once the data were synthesized.

In stage one of the thematic analysis, data on context and strategies initially collected were coded line-by-line inductively by three reviewers (GB, HC, LK). The coding process entailed constant comparison, ongoing development of new codes, and comparison with previous codes as each study was coded. Recommended best practices such as strong reviewers' engaged

**Table 1. Implementation outcomes adapted from Proctor et al. [34].**

| Outcome | Adapted definition |
|---|---|
| Appropriateness | Perceived fit, relevance, or compatibility of the innovation for a given setting or population |
| Feasibility | The extent to which an innovation can be successfully implemented in a given setting |
| Acceptability | A given service is agreeable, palatable, or satisfactory |
| Adoption | Intention, initial decision, or action to employ an innovation or practice, i.e., the uptake |
| Fidelity | The degree to which an intervention was implemented as it was prescribed and intended in the original protocol |
| Adaptation | Changes made to an intervention are based on deliberate considerations to increase fit with a patient or contextual factor |
| Penetration | Integration of practice within a service setting |
| Sustainability | The extent to which a newly implemented treatment is maintained or institutionalized within a service setting's ongoing operations |
| Implementation cost | The cost of an implementation effort |
| Scaling | Deliberate efforts to increase the impact of health service innovations to benefit more people and foster policy and program development on a lasting basis |
| Program outcomes | Observed effects of the program on domains of early childhood development |
| Impact | Long-term results of a program on early childhood development |

in data synthesis, reflexive analysis, and peer debriefing techniques were used to ensure methodological rigor throughout the process [42]. A code structure including (a) implementation contextual barriers and facilitators and (b) implementation strategies themes, was considered finalized at the point of theoretical saturation (when no new concepts emerged from the coding) [40, 42]. Likewise, implementation outcomes were coded line-by-line deductively to classify the content within the implementation outcomes framework defined by Proctor et al. [34]. All included records were then coded by three reviewers (GB, HC, LK) using a consistent coding process leading to a clear final code structure and codes' definitions (S3 Table) [41].

In stage two of the thematic analysis, three reviewers (GB, HC, LK) used an interpretative approach to compile data within and cross-site program into relationship codes to identify building blocks of implementation success. To ensure methodological rigor, multiple group discussions to develop a final consensus were held and any disagreements in relationship codes were solved by revisiting and further discussing the original open codes. This iterative process was repeated until the building blocks of implementation were deemed to be sufficient to describe the relationship between implementation context, strategies, and outcomes.

Lastly, contextual enabling and constraining factors that impacted the operation of implementation strategies and subsequent implementation outcomes were organized into building blocks of implementation success. To assure data collection quality and reliability of results, data on implementation strategies as well as implementation outcomes were recorded as present, not present, or not identified for each implementation site (S8 and S9 Tables). This evidence was used to revise the initial program theory that informed the development of a series of non-exhaustive, theory-based questions to guide the implementation planning of ECD interventions. All authors participated in consensus sessions to reach agreement on the final key conceptual framework, findings, and conclusions from the scoping review.

## Results

Our systematic search yielded 2,267 identified records, of which 75 records were eligible for data extraction (Fig 1 and S10 Table).

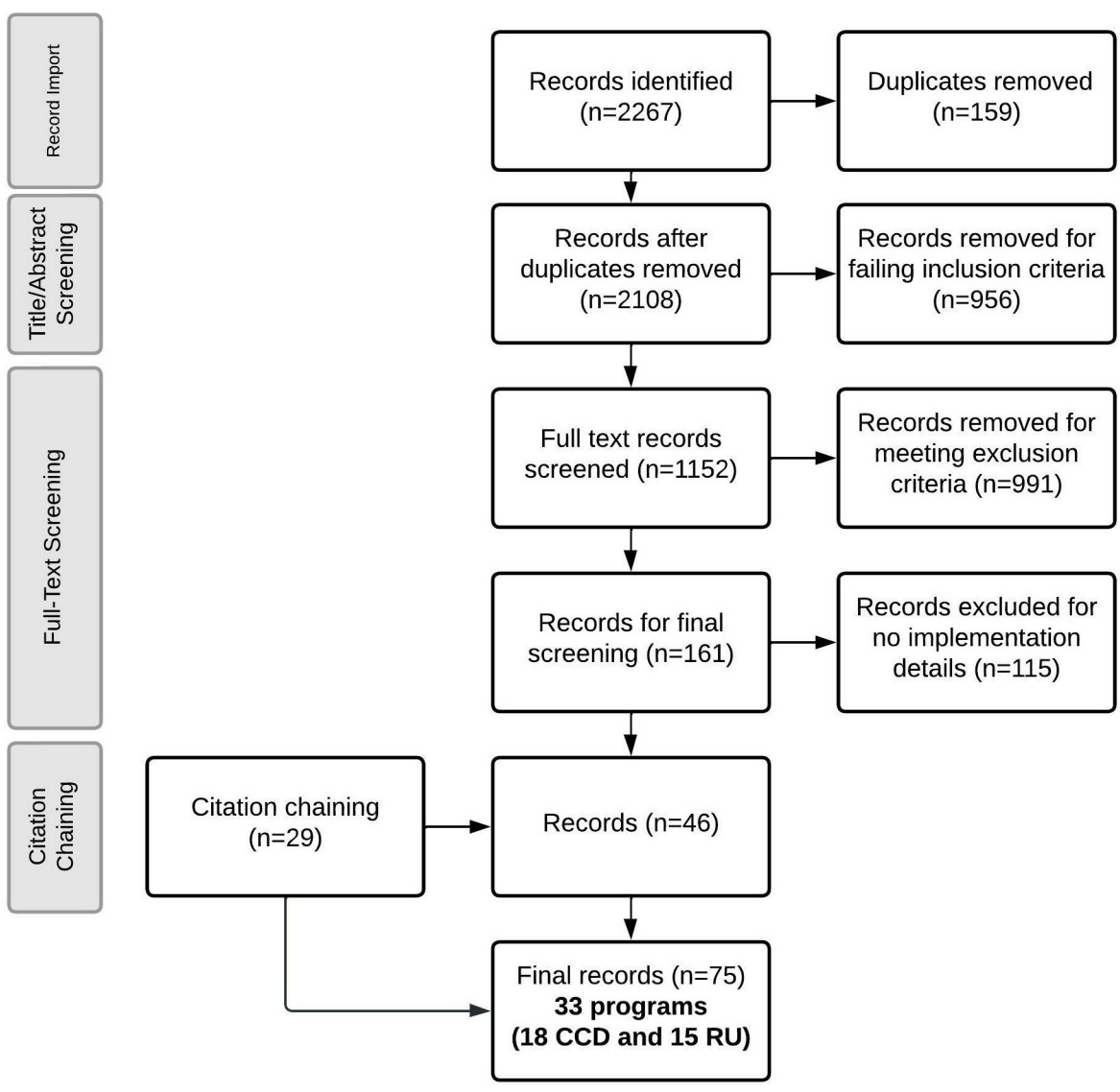

**Fig 1. PRISMA diagram.**

The records identified included implementation details on 33 programs across 23 countries. Eighteen CCD programs were implemented across 13 countries, and 15 RU programs across 10 countries (Fig 2). CCD was implemented across six world regions, including Africa (n = 6 programs) [9, 43–59], Americas (n = 1) [20, 60, 61], Eastern Mediterranean (n = 2) [9, 10, 27, 52, 62–67], Europe (n = 4) [8, 68], South-East Asia (n = 2) [9, 26, 69–74], Western Pacific (n = 3) [6, 9, 75–79]. RU was implemented in the Africa (n = 2) [80, 81], Americas (n = 5) [11–15, 37, 53, 80, 82–92], Eastern Mediterranean (n = 3) [93], and South-East Asia (n = 5) [37, 82, 94–101]. Brazil was identified as the only country that had implemented both CCD and RU.

The implementation context for each program is shown in Tables 2 and 3. The implementation of Jamaica's RU program began in 1986, and the European region started implementing CCD programs (n = 4) in the early 2000's with the majority of programs (n = 28) starting implementation after 2009. The vast majority of the programs were implemented in rural

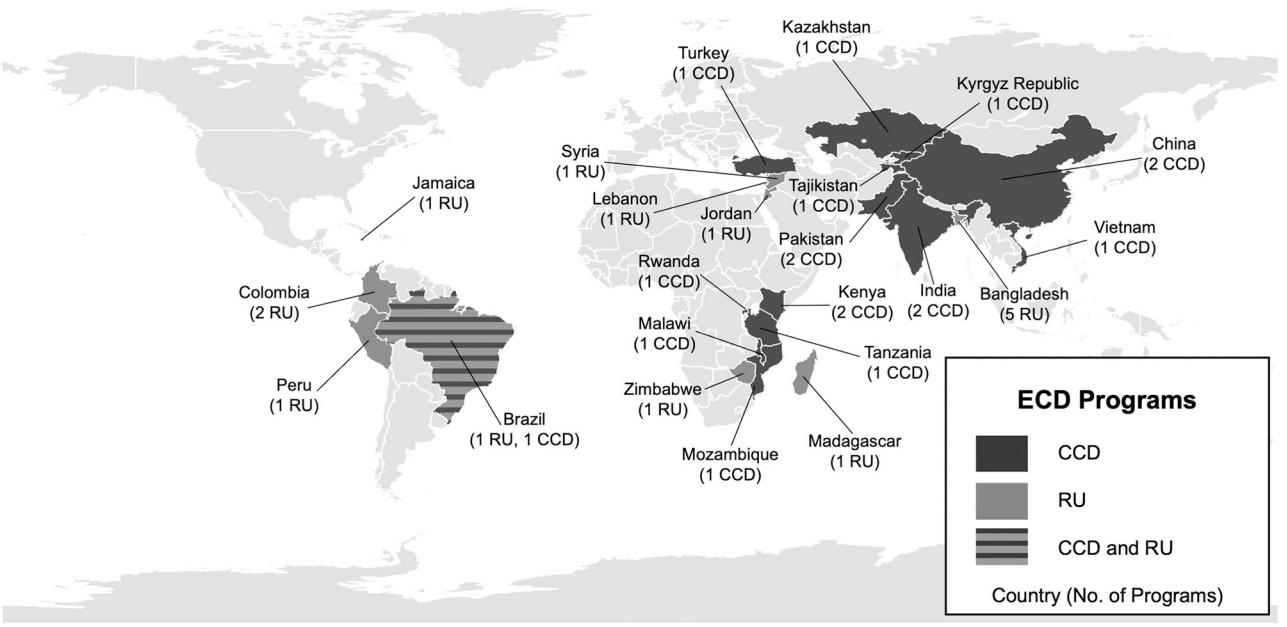

**Fig 2. Map of countries implementing Care for Child Development (CCD) and Reach Up (RU) globally.** Legend: Base geography layers retrieved from Natural Earth: https://www.naturalearthdata.com/downloads/10m-cultural-vectors/.

settings and consisted of feasibility trials lasting between 6 and 36 months. Among the CCD programs identified, Kenya and Brazil had ongoing programs at the time of our review, Brazil's Criança Feliz program had the largest reported reach both in rural and urban areas [20]. Among the RU programs, only Peru had an ongoing program, named Cuna Más, that began to be implemented since 2012 mostly in rural areas nationwide [83].

## Building blocks of implementation success

Six building blocks shaping RU and CCD implementation were identified: *program emergence*, *intersectoriality*, *intervention characteristics*, *workforce*, *training*, and *monitoring systems*. The number of implementation strategies' themes across building blocks ranged from five to eight (Table 4). Within the program emergence building block, thirty-two programs reported funding sources as an implementation strategy. The majority of programs were based on pilot research or feasibility trials. Intersectoriality was a component of the intervention characteristic in 76% (n = 25) of the programs. Within the intervention characteristics building block, 32 programs reported implementation strategies for intervention delivery, setting adaptations, and targeting. Within the workforce building blocks, workforce recruitment and training strategies were reported for 32 programs. Within the monitoring system building blocks, standardized data collection tools (n = 31) and supportive supervision (n = 25) were the most frequent implementation themes identified.

The three implementation outcomes found more frequently across building blocks were feasibility, appropriateness, and acceptability, respectively (Table 4). The three found less frequently were penetration, implementation cost, and adoption. As expected, the majority of the implementation outcomes reported were related to the intervention characteristics. Penetration and scaling outcomes were more frequently linked to the program emergence building block. Likewise, fidelity outcomes were linked to the monitoring system building blocks (Table 5).

**Table 2. Implementation context of each identified Care for Child Development (CCD) programs.**

| Country | Program Name | Years of Implementation (total duration) | Program Aims (implementation and/ or outcome goals of program) | Implementation Scale (Rural/ Urban) | Reach (n) | Child Age (in months) | Implementation Site |
|---|---|---|---|---|---|---|---|
| **African Region** | | | | | | | |
| **Kenya** (low income) | Msingi Bora | 2018 to 2019 (6 months) | **Implementation Goal:** Implement the ECD program with the most effective, cost-effective, and scalable delivery model in rural Kenya. **Outcome Goal:** Improve short- and medium-term child developmental outcomes by changing parental caregiving behavior. | Regional (rural) | 1070 children | 6–30 months | Study takes place in former Western and Nyanza Provinces in Kenya where there are high rates of poverty, child mortality, stunting ranging from 31–34%, and spousal violence, and adolescent motherhood. The area is very rural and most occupants work in subsistence farming or as unskilled informal workers. There is large linguistic diversity |
| | Smart Start Siaya County (PATH) | 2012 to present (ongoing) | **Implementation Goal:** Integrate CCD into the health system supported by a local government policy framework supporting ECD **Outcome Goal:** Improve ECD to support sustainable social and economic development | Regional (rural) | Not specified; county wide | 0–36 months | Siaya county is one of the most rural counties in the country. It has a high prevalence of HIV and under-5 mortality. Under five child mortality has been decreasing with the 2017 prevalence of 45.6 per 1,000 live births down by half since 2003. Stunting has fallen from 35% in 2008 to 26% in 2014. Challenges include teenage pregnancy, maternal mental health, father involvement, low access to pediatric medical and specialty care, unregistered children. Existing data suggests children are not meeting universal development standards with 50% of children in a neighboring county not on track to meet development benchmarks in three of four categories. |

*(Continued)*

**Table 2.** (Continued)

| Country | Program Name | Years of Implementation (total duration) | Program Aims (implementation and/or outcome goals of program) | Implementation Scale (Rural/Urban) | Reach (n) | Child Age (in months) | Implementation Site |
|---|---|---|---|---|---|---|---|
| **Malawi** (low income) | CCD | Formative qualitative assessment: 2012 to 2013 (7 months) Feasibility trial: 2013 to 2018 (duration not reported) | **Implementation Goal:** Adapt CCD to be culturally appropriate and acceptable to the existing community health worker workforce and caregivers. **Outcome Goal:** Improve ECD of children facing multiple insults of poverty, malnutrition, infection, and lack of stimulation. | Multi-Community (rural and urban) | 60 participants | 0–24 months | Implementation took place in the rural Mangochi district and the urban Blantyre district. Mangochi district is an area dependent on fishing and agriculture and has a mixture of Christian and Muslim community members. Blanture district is a major commercial city with a more diverse, mobile population. |
| **Mozambique** (low income) | Nurturing Care Collaboration (PATH) | 2018 to present (ongoing) | **Implementation Goal:** Strengthen the capacity of subnational health system actors to deliver ECD services for early learning and nutrition. **Outcome Goal:** Improved ECD and nutrition outcomes of children under 3 and increased families' awareness of nurturing care services | Regional (rural) | Not specified; universal in multiple counties | 0–36 months | This program was implemented in Monapo district. Monapo is a rural district with most locals working in agriculture in subsistence farming. Common ECD priorities and challenges included: low access to medical care, demanding workload of farming, malnutrition, stigma of developmental delays |
| **Rwanda** (low income) | Sugira Muryango | Pilot: 2014 to 2015 (duration not reported) Feasibility trial: 2018 (3–4 months) | **Implementation Goal:** Feasibly deliver the integrated ECD program with the use of community-based lay workers **Outcome Goal:** Support families living in poverty to improve caregiver child interactions, reduce family conflict, promote childhood development and ultimately break the intergenerational cycle of poverty. | Community/regional (rural) | Pilot 20 households; full trial 541 households | 6–36 months | Implemented in Rubona and Munyaga sectors of the Rwamagana district. |

(*Continued*)

**Table 2.** (Continued)

| Country | Program Name | Years of Implementation (total duration) | Program Aims (implementation and/or outcome goals of program) | Implementation Scale (Rural/Urban) | Reach (n) | Child Age (in months) | Implementation Site |
|---|---|---|---|---|---|---|---|
| **Tanzania** (low income) | Integrated health, nutrition, responsive stimulation package | Effectiveness trial: 2017–2019 (18 months) | **Implementation Goal:** Integrate home visit-based intervention, including responsive stimulation, health, and nutrition with and without conditional cash transfer program delivered by community health workers. **Outcome Goal:** Reduce pregnant and maternal depressive symptoms by addressing multiple risk factors and providing a range of coping strategies and peer support. | Regional (rural) | 593 households; 395 children under 12 months, and 198 pregnant women | 0–12 months and pregnant women | This program was implemented in 12 villages of Ifakara Health Institute HDSS in the Kilombero and Ulanga districts in the Morogoro region of Tanzania. The intervention area is predominately rural, and the majority of residents are subsistence farmers. Common ECD priorities included: 16.9% of low birthweight (< 2500 g) and 36.2% stunting (HAZ < -2) among children 18–36 months of age |
| **Americas Region** | | | | | | | |
| **Brazil** (middle income) | Criança Feliz | 2016 (ongoing) | **Implementation Goal:** Integrate the ECD home visiting program with an intersectoral nurturing care actions **Outcome Goal:** Teach parents in the most vulnerable communities how to provide opportunities for early learning by helping them develop their responsive parenting skills as a means of improving nurturing care to their children and ultimately seeks to help reduce poverty, inequities, and violence in the country | National (66.4% of eligible municipalities in Brazil—both rural and urban) | 13,000 children and 145,000 pregnant women | 0–36 months for all socially vulnerable children, and pregnant women, 0–72 months for children with disabilities | The program is implemented country-wide and it is one of the largest home visiting programs in the world. Young children under the age of 6 comprise 11% of the country's population of 200 million. Approximately 6.5% of all Brazilian families live below the poverty line, and nearly 25% of Brazilians are living in poverty. Roughly 42% of children under the age of 6 come from families whose income is below the poverty line. Brazil is a highly inequitable society which is captured through the great socio-economic variation across regions and the 5,570 municipalities. |
| **Eastern Mediterranean Region** | | | | | | | |

(*Continued*)

**Table 2.** (*Continued*)

| Country | Program Name | Years of Implementation (total duration) | Program Aims (implementation and/or outcome goals of program) | Implementation Scale (Rural/Urban) | Reach (n) | Child Age (in months) | Implementation Site |
|---|---|---|---|---|---|---|---|
| **Pakistan** (low-middle income) | Pakistan Early Child Development Scale-up Study (PEDs) | 2009 to 2012 (33 months) | **Implementation Goal:** Use Lady Health Worker program to scale up integrated nutrition and stimulation intervention delivery in a feasible, cost-effective, and effective way. **Outcome Goal:** Improve child development, growth and morbidity outcomes in rural Pakistan. | Regional (rural) | 1302 children | 0–24 months | In 2010, there were severe floods damaging homes and health facilities, challenging health outcomes and delivery. This intervention was implemented in the Nausehro Feroze district, a predominantly rural and impoverished district. |
| | Sustainable Program Incorporating Nutrition and Games (SPRING) | 2011 to 2016 (duration not reported) | **Implementation Goal:** Develop a feasible, affordable, and sustainable home visiting intervention using community-based workers. **Outcome Goal:** Improve maternal psychosocial wellbeing and child development. | Community (rural) | 37 families | 0–24 months | Program takes place in a low-income rural setting in Rawalpindi, Pakistan- Bagga Sheikhan Union Council with population 20,000. |
| **Europe Region** | | | | | | | |
| **Turkey** (upper-middle income) | Care for Development | 2004 (3 months) | **Implementation Goal:** Provide a cost-effective method of improving child development in a public healthcare setting with low resources **Outcome Goal:** Improve child development and address ECD disparities between high and low- and middle-income contexts | Community- one pediatric outpatient clinic (urban) | 120 children | 0–24 months | Program implemented in an outpatient pediatric clinic of a medical school in Ankara, a city of 4.5 million people. The clinic provides care to patients from low- or middle-income backgrounds. |
| **Kazakhstan** (low-middle income) **Tajikistan** (low income) **Kyrgyz Republic** (low income) | Better Parenting Initiative (Kazakhstan) Integrated Management of Childhood Illness with CCD (Tajikistan, Kyrgyz Republic) | Kazakhstan, Tajikistan: 2005 (duration not reported) Kyrgyz Republic: 2004 (duration not reported) | **Implementation Goal:** Integrate the integrated management of childhood illness and CCD into the national health system **Outcome Goals:** Strengthen parent's ability to support their children to improve child development and health. | Regional implementation across each of three countries (rural and urban) | Not reported | 0–36 months | Kazakhstan: Initiated in South Kazakhstan and expanded to East Kazakhstan in 2008 Tajikistan: Initiated in four districts, then expanded to health centers throughout the country Kyrgyz Republic: Trained CCD providers work in health centers across the country |
| **South-East Asia Region** | | | | | | | |

(*Continued*)

**Table 2.** (Continued)

| Country | Program Name | Years of Implementation (total duration) | Program Aims (implementation and/ or outcome goals of program) | Implementation Scale (Rural/ Urban) | Reach (n) | Child Age (in months) | Implementation Site |
|---|---|---|---|---|---|---|---|
| **India** (middle income) | Project Grow Smart | 2012 to 2013 (8 months) | **Implementation Goal:** Integrate a micronutrient program and early learning intervention to improve the development, grown, and nutrition of young children in rural India **Outcome Goal**: Improve the developmental, education, and economic potential of children | Regional (rural) | 513 infants | 6–12 months (children aged 12–48 months did not receive CCD but were part of the intervention) | The intervention took place in rural villages of Nalgonda District of Andrha Pradesh. Rates of infant anemia exceed 80% in this district. |
| | Sustainable Program Incorporating Nutrition and Games (SPRING) | 2011 to 2016 (60 months) | **Implementation Goal:** Develop a feasible, affordable, and sustainable intervention using community-based workers to deliver a home visiting program to improve early childhood growth and development **Outcome Goal:** Promote child development, grow and survival and maternal psychosocial wellbeing in rural India | Regional (rural) | 1726 participants | 0–24 months; women during pregnancy and post-partum periods | The intervention took place in Rewari district, Harwana State with a total population of 200,000 and is predominantly rural. The literacy rate in the state is 76% with female literacy being lower at 67%. There is a very low female to male infant sex ratio. Infant mortality is 41/1000 births which is around the national average. 46% of children under the age of five are stunted despite the province being considered to be "food secure." A sub-study of the population found that there are high rates of childhood adversity that are associated with poor growth and development. |

**Western Pacific Region**

(*Continued*)

**Table 2.** (Continued)

| Country | Program Name | Years of Implementation (total duration) | Program Aims (implementation and/or outcome goals of program) | Implementation Scale (Rural/Urban) | Reach (n) | Child Age (in months) | Implementation Site |
|---|---|---|---|---|---|---|---|
| **Vietnam** (low-middle income) | Learning Clubs | Pilot: 2014–2015 (duration not reported) RCT: 2016–2018 (18 months) | **Implementation Goal:** Create a psycho-educational ECD program in rural Vietnam that can be implemented universally. **Outcome Goal**: Improve physical and mental health of women and their children by addressing multiple risks to their well-being in a low-resource setting. | Community (rural) | 300 mothers and 100 fathers and grandparents | 0–24 months and pregnant women | Pregnant women in the study area of rural Vietnam are affected by household food insecurity, undernutrition, iodine deficiency, anemia, and intimate partner violence disproportionately compared to women in high-income countries. |
| **China** (low-middle, upper-middle income) | IMCI with CCD | 2003 (6 months) | **Implementation Goal:** Adapt Care for Development to be efficacious and appropriate in rural China. **Outcome Goal:** Improve psychological and physical child development through improved child caregiving practices. | Community (rural) | 50 families | 0–24 months | Program implemented in seven villages in a rural county with high poverty in Anhui Province. |
| | Integrated Early Childhood Development Program (IECD) | 2014–2016 (26 months*) | **Implementation Goals:** To effectively deliver a comprehensive nurturing care intervention to children in poor, rural areas. **Outcome Goal**: Reduce neurodevelopmental child delays to promote subsequent educational achievement and adult productivity. | Regional (rural) | 5698 participants (N = 2953 children under 36 months and their caregivers were enrolled at baseline; N = 2745 child-caregiver pairs completed the postintervention assessment) | 0–36 months | Program implemented in four rural counties with high poverty: Songtau and Liping counties in Guizhous province and Fenxi and in counties in Shanxi Provinces. |

In the following sections we describe the context, implementation strategy, and outcomes that resulted from the thematic analysis synthesis.

**Building block 1: Program emergence.** The first building block identified was *program emergence*, which refers to the critical decisions for program design and adaptation made before implementation (Table 6). During program emergence, implementers made key implementation decisions regarding the remaining building blocks. The main <u>contextual</u> barriers to program emergence were lack of infrastructure and poor leadership and communication during the planning phase. On the other hand, the presence of local advocates who championed the initiative influenced the community motivation and trust to strengthen and support the program emergence (see Table 7 for a country-specific example). Five <u>implementation strategy themes</u> were identified: Governance and leadership (n = 24 programs, e.g. identification of early advocates through formal and informal ECD policy), Multilevel Networks (n = 28, e.g.

**Table 3. Implementation context of each identified Reach Up (RU) programs.**

| Country | Program Name | Years of Implementation (total duration) | Program Aims (implementation and/or outcome goals of program) | Implementation Scale (Rural/Urban) | Reach (n) | Child Age (in months) | Implementation Site |
|---|---|---|---|---|---|---|---|
| **African Region** | | | | | | | |
| **Zimbabwe** (low income) | Modified Reach Up | 2015–2018 (36 months) | **Implementation Goal:** Not reported **Outcome Goal:** Improve parenting skills of responsive caregiving in order to improve children's child development gains. | Community (rural) | 200 children | 6–48 months | Not reported |
| **Madagascar** (low income) | Early Stimulation | 2014–2016 (24 months) | **Implementation Goal:** Determine whether nutrition supplementation, stimulation, or a combined intervention would best address severe stunting and/or ECD delays among young children in Madagascar. **Outcome Goal:** Use an innovated integrated nutrition and caregiving program to reduce chronic malnutrition and promote child development in low-income settings. | Community (rural/ urban) | 2490 children | 0–24 months | Program sites were sampled from five regions in South and Southeast Madagascar: Amoron'i Mania, Androy, Atsimo Atsinanana, Haute Matsiatra, and Vatovavy-Fitovinany. These regions have some of the highest prevalence of child stunting in the nation. Specifically, these sites were selected for World Bank emergency loans to restore and strengthen basic health service delivery following the political and economic crisis between 2009 and 2012. |
| **Americas Region** | | | | | | | |
| **Jamaica** (low income) | Jamaican Home Visiting Program | 1986–1989 (24-month intervention with additional month for study activities) | **Implementation Goal:** Development of a novel, effective ECD program that improves child development and associated characteristics in low-resource communities **Outcome Goal:** Improve child development of children in low-resource households. | Community (urban/peri-urban) | 129 children | 9 to 24 months | The program was conducted in poor neighborhoods of the city of Kingston and adjoining parishes of St. Andrew and St. Catherine. |
| **Colombia** (middle income) | Home-based Early Childhood Development Intervention | 2010–2013 (18 months) | **Implementation Goal:** Not identified **Outcome Goal:** Improve parenting practices and enhance psychosocial stimulation in young children, both of which benefit ECD outcomes to promote long-term health. | Regional (rural) | 1,419 children | 12–42 months | The intervention was targeted at families receiving the Colombian CCT program, and was implemented in semi-urban municipalities in 3 regions of central Colombia. |
| | Enhanced FAMI | 2014–2016 (average duration 11 months, but duration varied by implementing community) | **Implementation Goal:** Improve quality of parenting program with a nutrition component implemented at scale. **Outcome Goal:** Improve child cognitive development, decrease risk stunning, and enhance early learning environment. | Regional (rural) | 2,134 children | 0–24 months | Implemented in central rural and peri-rural regions: Boyaca, Cundinamarca, Santander, Tolima |
| **Brazil** (middle income) | Responsive Caregiving and Early Learning Program | 2015–2016 (12 months) | **Implementation Goal:** Evaluate efficacy and cost-effectiveness of the home visiting program. **Outcome Goal:** Improve parenting skills of responsive caregiving in order to improve children's child development gains. | Community (urban) | 400 children | 6–48 months | This intervention was implemented in western urban districts of Sao Paulo, the largest city in Brazil. Sao Paolo contains slums and over 30% of the population receive less than half the federal minimum wage. |
| **Peru** (middle income) | Cuna Más | 2012-present (on-going) | **Implementation Goal:** Not identified **Outcome Goal:** Improve early childhood cognition, language, physical and socioemotional development, and improve family knowledges and practices to strengthen attachment. | National (rural) | 149,000 children | 0–36 months | Cuna Más is aimed at children living in poverty across rural and urban communities. In marginalized urban areas, a daycare service is available for children age 6–36 months. In rural communities, home visiting services and monthly group sessions for children under 36 months and their primary caregivers and pregnant women are offered. |
| **Eastern Mediterranean** | | | | | | | |
| **Syria, Lebanon, Jordan** (low income) | Modified Reach Up and Learn | 2017–2019 (52 months) | **Implementation Goal:** Creating environments that support engagement and opportunities for kids ages 0–3 to achieve optimal ECD outcomes. **Outcome Goals:** Not identified | Community (rural) | 4,089 children | 6–42 months | A mapping of sectors and program statuses was conducted to inform which sectors would implement in Jordan and Lebanon. In Lebanon, intervention was implemented in peri-urban areas. In Jordan, it was implemented in peri-urban areas and informal tented settlements. In Syria, it was implemented in the Northeast region in peri-urban and camps for internally displaced persons. |
| **South-East Asia Region** | | | | | | | |

(Continued)

**Table 3.** (Continued)

| Country | Program Name | Years of Implementation (total duration) | Program Aims (implementation and/or outcome goals of program) | Implementation Scale (Rural/Urban) | Reach (n) | Child Age (in months) | Implementation Site |
|---|---|---|---|---|---|---|---|
| **Bangladesh** (low income) | Integrated psychosocial stimulation and unconditional cash transfer | July 2017—December 2018 | **Implementation Goal:** Integrated psychosocial stimulation intervention to lactating mothers enrolled in an unconditional cash transfer. **Outcome Goal:** Improve the child development. | Community (rural) | 594 children | 6–16 months | The program was implemented in Ullapara subdistrict because it had sufficient number of wards (a clearly demarked area) for cluster randomization. Also, it is located 180km from Dhaka facilitating regular visits by the research staff based in Dhaka. |
| | Integrated responsive stimulation, maternal mental health, nutrition, WASH and lead exposure prevention interventions (RINEW) | March- April 2017 (Pilot) and September 2017- May 2018 (Trial) | **Implementation Goal:** Integrate a psychosocial stimulation, maternal mental health, nutrition, WASH and lead exposure prevention interventions targeting pregnant and lactating mothers. **Outcome Goal:** Improve caregiving practices, child development, and caregiver mental health. | Community (rural) | 40 participants (pilot) 621 participants (trial) | 0–24 months (pilot) 0–15 months (trial) | The pilot program was implemented by Community Health Works in two villages (Adampur and Chorbetal) in Kishoreganj district. The trial program was implemented by Community Health Works in villages with population between 200 to 800 households located in Katiadi and Kuliarchar subdistricts of Kishoreganj district. |
| | Modified Reach Up | 2014–2016 (12-month intervention phased in by districts) | **Implementation Goal:** Integrate the intervention into existing routine government health services for underweight children from poor families. **Outcome Goal:** Improve the child development of malnourished children. | Community (rural) | 1,597 children | 5–24 months | The program was implemented at community clinics in Narsingdi district. This district was selected because it had a sufficient number of community clinics and was less than 80 km from the city of Dhaka. |
| | Psychosocial Stimulation | Not reported (6 months) | **Implementation Goal:** Evaluate effectiveness of combined and independent nutrition and responsive caregiving interventions on child nutrition and development. **Outcome Goal:** Prevent severe developmental delays among malnourished children. | Community (urban) | 507 children | 6–24 months | This intervention was implemented in a centrally-located hospital and follow-up nutrition clinics in four urban slums of Dhaka city. |
| | Modified Jamaican Home Visiting Program | 2000–2002 (12 months program implementation, additional months preparing and measuring follow-up data) | **Implementation Goal:** Compare mechanism of combined and independent nutrition and responsive caregiving interventions on child nutrition and development. **Outcome Goal:** Improve ECD outcomes of undernourished Bangladeshi children. | Community (peri-urban) | 313 children | 6–24 months | This program was implemented in the Monohardi subdistrict which is about a two-hour drive outside the large city of Dhaka. This is a rural area with high poverty and most residents are farmers. |

joint planning among multisectoral partners), Funding (n = 32, e.g. use of local, state, and/or federal funding sources), Scaling Process (n = 20, e.g. conducting a feasibility trial before scaling to allow implementers to adapt the program), Integration into ECD system (n = 25, e.g. government ownership allows integration of service provided by the social and health systems). Key program design decisions during program emergence facilitated implementation outcomes such as scaling (n = 13), adaptation (n = 1), appropriateness (n = 5), and adoption (n = 5). An appropriate implementation progression throughout program emergence was shown to promote feasibility (n = 9), acceptability (n = 1), fidelity (n = 1), penetration (n = 4), and sustainability (n = 5). Implementation cost (n = 1) were not commonly found within program emergence.

**Building block 2: Intersectoriality.** The second building block identified was *intersectoriality*, which refers to the collaborations between stakeholders and institutions representing diverse sectors related to ECD (i.e., ministries of health and education, NGO partners, university-based academics) to support program implementation (Table 8). The identified contextual barriers to promoting intersectoriality were an overload of existing ECD-related services, workforce competing priorities, and challenges to coordinate referrals to multiple services required to address families' multiple needs. The presence of an existing network of scaled family services enabled programs to combine new ECD programming and pre-existing delivery mechanisms with a diverse workforce already in place (see Table 9 for a country-specific example). Five implementation strategy themes were identified: Coordination and Communication (n = 14, e.g., define level of commitment, roles, and scope of work among

**Table 4. Implementation strategies' themes identified in the thematic analysis by building blocks of implementation of Early Childhood Development Programs.**

| Building blocks of implementation | Implementation Strategies Themes | | Programs reporting | |
|---|---|---|---|---|
| | | | n | % |
| Program Emergence | IS1.1 | Governance and Leadership | 24 | 73% |
| | IS1.2 | Multilevel Networks | 28 | 85% |
| | IS1.3 | Funding | 32 | 97% |
| | IS1.4 | Scaling Process | 20 | 61% |
| | IS1.5 | Integration into ECD system | 25 | 76% |
| Intersectoriality | IS2.1 | Coordination and Communication | 14 | 42% |
| | IS2.2 | Intervention Characteristics | 25 | 76% |
| | IS2.3 | Intersectoral Targeting | 12 | 36% |
| | IS2.4 | Intersectoral Workforce | 18 | 55% |
| | IS2.5 | Delivery Sites and Networks | 15 | 45% |
| Intervention Characteristics | IS3.1 | Intervention Delivery | 32 | 97% |
| | IS3.2 | Intervention Setting | 32 | 97% |
| | IS3.3 | Dose and Frequency | 24 | 73% |
| | IS3.4 | Design Approach | 32 | 97% |
| | IS3.5 | Materials | 30 | 91% |
| | IS3.6 | Caregiver Engagement | 15 | 45% |
| | IS3.7 | Supplementary program activities | 21 | 64% |
| | IS3.8 | Responsive Teaching | 15 | 45% |
| Workforce | IS4.1 | Workforce recruitment | 32 | 97% |
| | IS4.2 | Implementation Agent Training | 32 | 97% |
| | IS4.3 | Job Satisfaction | 11 | 33% |
| | IS4.4 | Compensation and Contracts | 16 | 48% |
| | IS4.5 | Caseload | 13 | 39% |
| | IS4.6 | Workforce Characteristics | 21 | 64% |
| | IS4.7 | Community Integration | 20 | 61% |
| Training | IS5.1 | Training manuals and protocols | 23 | 70% |
| | IS5.2 | Training Skills | 17 | 52% |
| | IS5.3 | Continuing Education | 17 | 52% |
| | IS5.4 | Training Leadership | 16 | 48% |
| | IS5.5 | Training Design | 29 | 88% |
| | IS5.6 | Institutionalization | 7 | 21% |
| Monitoring System | IS6.1 | Supportive Mentoring | 25 | 76% |
| | IS6.2 | Standardized Data Collection Tools | 31 | 94% |
| | IS6.3 | Data-Informed Decision Making | 20 | 61% |
| | IS6.4 | Supervisor Training | 18 | 55% |
| | IS6.5 | Supervisor Caseload | 11 | 33% |
| | IS6.6 | Impact Measurement | 30 | 91% |

collaborators), Intervention Characteristics (n = 25, e.g., address multiple domains of child health, nutrition, and development), Intersectoral targeting (n = 12, e.g. use structure of existing scaled up programs), Intersectoral Workforce (n = 18, e.g., use of existing workforce jobs or tasks), and Delivery Sites and Networks (n = 15, e.g., use a well-established health system network at the community level). The use of existing ECD networks and supervisory capacity were found to promote <u>implementation outcomes</u> such as program acceptability (n = 4),

**Table 5. Implementation outcomes reported by building blocks of implementation of Early Childhood Development Programs.**

| Implementation outcomes | # of programs reporting | | | | | | | | | | | | # of times implementation outcome was reported |
|---|---|---|---|---|---|---|---|---|---|---|---|---|---|
| | Program Emergence | | Intersectoriality | | Intervention Characteristics | | Workforce | | Training | | Monitoring System | | |
| | n | % | n | % | n | % | n | % | n | % | n | % | |
| Appropriateness | 5 | 9% | 3 | 5% | 24 | 44% | 6 | 11% | 12 | 22% | 5 | 9% | 55 |
| Feasibility | 9 | 15% | 6 | 10% | 16 | 26% | 13 | 21% | 11 | 18% | 7 | 11% | 62 |
| Acceptability | 1 | 2% | 2 | 4% | 20 | 42% | 13 | 27% | 11 | 23% | 1 | 2% | 48 |
| Adoption | 5 | 20% | 3 | 12% | 6 | 24% | 7 | 28% | 3 | 12% | 1 | 4% | 25 |
| Fidelity | 1 | 3% | 1 | 3% | 7 | 18% | 7 | 18% | 5 | 13% | 17 | 45% | 38 |
| Adaptation | 1 | 3% | 1 | 3% | 25 | 69% | 0 | 0% | 7 | 19% | 2 | 6% | 36 |
| Penetration | 4 | 36% | 2 | 18% | 4 | 36% | 1 | 9% | 0 | 0% | 0 | 0% | 11 |
| Sustainability | 5 | 13% | 4 | 11% | 8 | 21% | 6 | 16% | 9 | 24% | 6 | 16% | 38 |
| Implementation Cost | 1 | 5% | 1 | 5% | 8 | 42% | 6 | 32% | 3 | 16% | 0 | 0% | 19 |
| Scaling | 13 | 39% | 5 | 15% | 4 | 12% | 3 | 9% | 5 | 15% | 3 | 9% | 33 |

**Table 6. Program emergence building block of Care for Child Development (CCD) and Reach Up (RU) implementation pathways.**

| Context | | Implementation Strategies (Number of programs reporting) | | | Implementation Outcomes (number of programs reporting) | |
|---|---|---|---|---|---|---|
| Facilitators | Barriers | | | | | |
| ECD Disparities: Gaps in ECD services and resources widen inequities in children's developmental outcomes; few governmental sectors reach the nation's youngest children. | Capacity: Some ECD sectors may be overloaded at the time of implementation while others may have infrastructure that can be scaffolded. | IS1.1 | Program Governance and Leadership: Individuals and groups with regional and/or national influence who are early advocates of CCD or RU can help drive early program support and sustain program activities. This can be achieved through formal and informal ECD policy, advocacy, and issue awareness-raising. | (24) | Appropriateness | (5) |
| | | | | | Feasibility | (9) |
| | | | | | Acceptability | (1) |
| | | | | | Adoption | (5) |
| | | | | | Fidelity | (1) |
| | | | | | Adaptation | (1) |
| | | | | | Penetration | (4) |
| | | | | | Sustainability | (5) |
| ECD System (Legislation): The presence of an existing public infrastructure providing services to children and families. ECD support is codified in legislation as a policy commitment. | Communication: If program leaders do not have formal platforms to communicate with one another there is a risk of ineffective leadership and poor planning. | IS1.2 | Multi-Level Networks: Joint planning among inter-and intra-sectoral partners, including community leaders and members, is crucial to implementation and planning. Having a local community champion supports acceptance and local buy-in. | (28) | Implementation Cost | (1) |
| | | | | | Scaling | (13) |
| Community: Community interest in improving child health and development drives adoption. | | IS1.3 | Funding: The use of local, state, and/or federal funding sources ensures sustainability (as opposed to the unpredictability of funding from external organizations) and ensures the potential for implementation with fidelity and scaling. Continued and expanded program funding can be dependent on evidence of program effectiveness and implementation cost analyses collected via monitoring systems. | (32) | | |
| Curriculum: CCD and RU are evidence-based methodologies for responsive caregiving interventions promoted by UNICEF/WHO and international partners. | | IS1.4 | Scaling Process: Conducting a feasibility trial before scaling allows implementers to adapt the program. It may be necessary to take pauses in program scaling up to ensure the quality of the program. Scaling may be regional or national. | (20) | | |
| | | IS1.5 | ECD System Integration: Government ownership of the program from the beginning allows for the integration of the program with social services or the health system using existing workforces. | (25) | | |

**Table 7. Program emergence: Kenya, Smart Start (Care for Child Development).**

Smart Start is an ECD initiative based in Kenya's Siaya County. Powered by strong central leadership, community buy-in, and supportive ECD policies, the program was initiated in 2012 following legislation in 2006 and 2010 that established national ECD policy and frameworks to be implemented through the decentralized county governments. The Siaya County governor and Kenyan first lady led a coordinated county-wide effort to support local ECD. In partnership with university ECD centers and foundations, CCD home visits were integrated with other ECD efforts to form the Smart Start program in 2014. A high priority was placed on embedding ECD activities into all local public services including home garden programs addressing child nutrition, hygiene and sanitation village outreach for preventing child diarrheal disease, and local reforms of the justice system to improve its capacity to prevent and mitigate child abuse and neglect. The presence of strong central leadership and use of multisectoral activities led to high program visibility and uptake by clinics' staff. Village leaders served as community champions of Smart Start, further encouraging families to participate. Central leadership from the governor and first lady was informed by data collected through a multisectoral team, which coordinated intra-sectoral partnerships and monitoring activities to capture data on program successes and challenges. Following one year of implementing home visits, the absence of government stipends for the community health workers delivering Smart Start home visits and activities was identified as a major barrier to program success. In response, Siaya County became the first Kenyan County to pass legislation guaranteeing government stipends for community health volunteers. This robust, multisectoral program was feasible, sustainable, and successfully implemented at full scale within Siaya County due to the focused leadership of the local government and ongoing collaborations with diverse multi-level stakeholders.

fidelity (n = 1), and penetration (n = 2). Collaborations with other sectors were found to strengthen the appropriateness for service providers and caregivers (n = 3) and feasibility (n = 6). Intersectoral partnerships established among the community and/or national leadership further facilitated the sharing of goals and adoption (n = 3), penetration by integrating delivery plans among key stakeholders (n = 2), reducing implementation cost or cost-effectiveness (n = 1), promoting sustainability (n = 4), and scaling (n = 5).

**Building block 3: Intervention characteristics.** The third building block identified was *intervention characteristics*, which refers to the use of a comprehensive curriculum and the specifications of its delivery (Table 10). The Contextual barriers identified related to time, geography and resources, led to the adaptations of diverse intervention characteristic such as changing location of sessions and using literacy and cultural appropriate toys (see Table 11 for a country-specific example). Eight implementation strategy themes were identified: Intervention delivery (n = 32, e.g. define types of delivery sessions), Intervention Setting (n = 32, e.g. select accessible location to deliver the program), Dose and Frequency (n = 24, e.g. define dose and frequency according to the needs and the setting), Design approach (n = 32, e.g. define selective or indicated targeting using equitable criteria), Materials (n = 30, e.g. adapt culturally sensitive written and visual materials), Caregiver engagement (n = 15, e.g. define recruitment strategies for diverse caregivers), Supplementary program activities (n = 21, e.g. add other domains of nurturing care to the intervention), Responsive teaching (n = 15, e.g. define strategies to engage caregivers during sessions). Well-defined yet adaptive and flexible intervention characteristics may lead to improved implementation outcomes including appropriateness (n = 24), feasibility (n = 16), acceptability (n = 20), adoption (n = 6), fidelity (n = 7), adaptation (n = 25), penetration (n = 4), sustainability (n = 8), implementation cost (n = 8), and scaling (n = 4).

**Building block 4: Workforce.** The fourth building block identified was *workforce*, which refers to the people delivering the intervention. The ability of the implementation agent to build a positive relationship with the caregiver is crucial to intervention acceptability and program success (Table 12). The Contextual barriers found were lack of motivation, overworked/overburdened workforce as well as low workforce compensation, which is one of the largest drivers of cost and closely tied to workforce retention (see Table 13 for a country-specific example). Seven implementation strategy themes were identified: Workforce Recruitment

**Table 8. The intersectoriality building block of Care for Child Development (CCD) and Reach Up (RU) implementation pathways.**

| Context | | Implementation Strategies (Number of programs reporting) | | | Implementation Outcomes (number of programs reporting) | |
|---|---|---|---|---|---|---|
| **Facilitators** | **Barriers** | | | | | |
| Program Governance and Leadership: Establishing clear leadership roles and responsibilities within and between sectors at multiple implementation levels is crucial. Messaging among and from these entities must be consistent and compelling to drive program acceptability. Sectors are willing to create common goals and work together. | Workforce: Intersectoral workforce has existing responsibilities sometimes making it difficult to integrate additional CCD or RU activities | IS2.1 | Coordination and Communication: There is a need for clear commitments, roles, and scope of work to be laid out among collaborators to avoid confusion and inefficiencies. This can allow space for intentional adaptation to local contexts and consideration of stakeholder capacities. When relevant, referrals can be easily made from RU or CCD implementers to additional health or social service. | (14) | Appropriateness | (3) |
| | | | | | Feasibility | (6) |
| | | | | | Acceptability | (2) |
| | | | | | Adoption | (3) |
| | | | | | Fidelity | (1) |
| | | | | | Adaptation | (1) |
| | | | | | Penetration | (2) |
| | | | | | Sustainability | (4) |
| | | | | | Implementation Cost | (1) |
| | | | | | Scaling | (5) |
| ECD System (Existing Services): The existence of established, accessible social services (i.e., healthcare, early education, conditional cash transfer program) that address a broad array of problems and support children and caregivers is beneficial to effective intersectoral collaborations. | Referrals: To address the family's multiple needs, CCD or RU implementation agents must make referrals to other services; however, this is difficult if there is not a strong network of services in place. | IS2.2 | Intervention Characteristics: Program delivery along with other interventions that support child health and development (e.g., nutritional supplements) addresses multiple child health and development threats concurrently. | (25) | | |
| Institutional Integration of ECD: From federal policy to local government, ECD is valued and recognized as an important aspect of health care (e.g., all medical students in the country are trained on ECD). | Leadership: The group who should be responsible for ECD based on formal responsibilities may not have the capacity to implement RU or CCD or may not be interested, therefore other groups may need to step in. | IS2.3 | Intersectoral Targeting: The use of existing, scaled conditional cash transfer programs or other services that can identify vulnerable families can help program reach and enrollment; however, these systems should not present further bureaucratic barriers for families in need. | (12) | | |
| Demand: Families and caregivers welcome user-friendly services, such as new programs that are integrated into existing services. | | IS2.4 | Intersectoral Workforce: Existing cadres of health workers and providers, educators, or NGO staff can integrate program delivery within their existing jobs or tasks. | (18) | | |
| | | IS2.5 | Delivery Sites and Networks: RU or CCD can use existing facilities to serve as familiar sites for program delivery. A well-established health system network at the community level can be used. Other well-known community sites such as schools or community meeting halls can also serve the same purpose if available. | (15) | | |

(n = 32, e.g. establish a process to hire local community members), Implementation Agent Training (n = 29, e.g. develop culturally-sensitive curriculum materials), Job Satisfaction (n = 11, e.g. access to easy-to-use tools and guidelines), Compensation and Contracts (n = 16, e.g. adequate compensation or stipends for program delivery), Caseload (n = 13, e.g. manageable caseload considering the catchment the geography of the catchment area), Workforce Characteristics (n = 21, e.g. level of education of implementation agents), Community Integration (n = 20, e.g. use of local workforce familiar with the caregivers in the community). Workforce implementation outcomes such as acceptability (n = 13), feasibility (n = 13), adoption (n = 7), fidelity (n = 7), appropriateness (n = 6) resulted from key decisions related to intervention characteristics. Use of existing workforce facilitated penetration (n = 1), scaling (n = 3), sustainability (n = 6), and reduced implementation cost (n = 6).

**Building block 5: Training.** The fifth building block identified was the *training* of the implementation agents, or the people working as home visitors or who deliver CCD or RU in a public or group setting outside of family homes (Table 14). Training of implementation agents is dependent on the existing ECD infrastructure, including the workforce and their

**Table 9. Intersectoriality: Pakistan, Early Child Development Scale-up Study (Care for Child Development).**

The Pakistan Early Child Development Scale-up Study (PEDS), implemented from 2009 to 2012, is an example of an intersectoral ECD program using existing child health infrastructure to deliver a responsive caregiving intervention. The CCD curriculum was successfully integrated into a nutrition intervention delivered by the well-established national Lady Health Worker Program. With a workforce of 100,000 female community health workers, the Lady Health Workers provided needed health outreach, including community and home visits, to supplement areas with weaker and rural clinical infrastructure. Local and international researchers partnered with the Lady Health Worker Program to provide additional training for the PEDS intervention. Lady Health Supervisors also received additional training to adequately supervise these new activities. The program delivered responsive caregiving coaching to parents in combination with nutrition education and micronutrient powder distribution. The existing Lady Health Worker workforce, which was coordinated by health authorities, served as an effective platform to deliver additional responsive caregiving and nutrition outreach activities to vulnerable rural families. Importantly, Lady Health Workers only had time to complete these additional activities because most health workers were not adequately meeting their pre-existing work requirements. When integrating responsive caregiving programs within existing health outreach programs, it is critical to monitor intervention delivery quality and not overburden the implementation agents.

responsibilities, and the local environment. The main barriers found were external pressure to scale and challenge to maintaining the training cascade that involves external actors in the training leadership (as opposed to local leaders). Training cascades with a clear delineation of job roles and responsibilities were most successful when led by local leaders in collaboration with experts (see Table 15 for a country-specific example). Six implementation strategy themes were identified: Training Manuals and Protocols (n = 23, e.g. standardize evidence-based manuals), Training Skills (n = 17, e.g. assess of skills and behaviors to deliver the program), Continuing Education (n = 17, e.g. introduce new content and scaffold upon existing skills), Training Leadership (n = 16, e.g. use of training cascade), Training Design (n = 29, e.g. define training delivery mode and length), Institutionalization (n = 7, e.g. embed training into existing ECD services and workforce development programs). The ability to adapt (n = 7) the training to contextual and skills needs improved implementation outcomes including appropriateness (n = 12), feasibility (n = 11), acceptability (n = 11), and adoption (n = 3). Institutionalization of refresher training strategies such as online professional development, and continuing education were implementation strategies found to be effective to increase fidelity (n = 5), sustainability (n = 9), implementation cost (n = 3), and scaling (n = 5).

**Building block 6: Monitoring system.** The sixth building block was the program *monitoring system*, the purpose of which was to provide a clear, reliable mechanism for measuring and evaluating program activities (Table 16). Key to the success of a monitoring system is the inclusion of supervisors which in turn facilitate program scale up. Consistent, timely mentorship and feedback from supervisors to implementation agents enable program outcomes to be directly monitored and improved, which maintains and strengthens program quality (see Table 17 for a country-specific example). The main contextual barriers found were logistical challenges to covering a wide geographic area and lack of infrastructure. Six implementation strategy themes were identified: Supportive Supervision (n = 25, e.g. define mode of delivering), Standardized Data Collection Tools (n = 31, e.g. use of electronic monitoring systems to store program data), Data-Informed Decision Making (n = 20, e.g. evidence-based programmatic decisions), Supervisor Training (n = 18, e.g. define intensity of supervisor training), Supervisor Caseload (n = 11, e.g. define supervisor responsibilities based on a manageable caseload), and Impact Measurement (n = 30, e.g. integrate child development indicators into existing national health surveillance system). Supportive Supervision favored implementation outcomes including adoption (n = 1), appropriateness (n = 5), feasibility (n = 7), fidelity

**Table 10. Intervention characteristics building block of Care for Child Development (CCD) and Reach Up (RU) implementation pathways.**

| Context | | Implementation Strategies (Number of programs reporting) | | Implementation Outcomes (number of programs reporting) | |
|---|---|---|---|---|---|
| **Facilitators** | **Barriers** | | | | |
| ECD System (Delivery): Facilities, where children and families have pre-existing and ongoing touchpoints with ECD services, can allow for the scaffolding of programs onto existing delivery mechanisms (e.g., pediatric visits, conditional cash transfer program services) and promote higher dose delivery. If no such systems are well-established, delivery through novel platforms may be most appropriate. | Feasibility: Limitations related to time, geography, and resources may make it unfeasible to provide individual sessions to participating families. Instead, group sessions may be provided. | IS3.1 | Intervention Delivery: We identified three forms of program delivery: (1) individual sessions in which implementation agents deliver the program to one family, (2) group sessions in which implementation agents deliver the program to multiple families at once, (3) hybrid sessions in which both individual and group sessions are used intermittently. | (32) | Appropriateness (24)<br>Feasibility (16)<br>Acceptability (20)<br>Adoption (6)<br>Fidelity (7)<br>Adaptation (25)<br>Penetration (4)<br>Sustainability (8)<br>Implementation Cost (8)<br>Scaling (4) |
| Infrastructure: Infrastructure capacity, including roads and travel time, electricity, heat, and technical capacity/preference, must be accounted for. | Location of Sessions: Choosing between delivering the program at families' homes and public settings demands optimizing the tradeoff between convenience for families and implementation agents. | IS3.2 | Intervention Setting: The intervention is delivered in a location accessible to both participating families and implementation agents. The intervention can be delivered in the homes of participating families or in accessible public settings such as local schools, churches, houses of village leaders, or community meeting halls. | (32) | |
| Population Characteristics: Underlying literacy level, socioeconomic level, cultural/religious practices, and gender norms of the target community shape program implementation. | Caregiver Engagement: Underlying social norms and economic constraints may prevent caregivers other than mothers (i.e., fathers) from participating. | IS3.3 | Dose and Frequency: Consistent delivery of program sessions, with potential booster sessions, supports sustained program effect and caregiver behavior adoption. Monthly visits were found to be less effective than weekly or fortnightly visits, but more feasible in resource-constrained settings if delivered with high quality and fidelity. Dose and frequency must be adapted to the setting. | (24) | |
| | Learning Retention: Caregivers may have a difficult time remembering RU or CCD content and suggested behaviors over time; this could be addressed by booster follow-up sessions. | IS3.4 | Designed approach: Delivery of the intervention to children as early as possible will result in the biggest effects on ECD. Selective or indicated targeting facilitates efficient use of program funding to support the most vulnerable children. It is important that the intervention targeting is transparent and equitable. | (32) | |
| | Materials: Typically, caregivers preferred having materials such as toys left at their house between sessions so they could practice with their children; however, this was not always feasible due to program resource constraints. | IS3.5 | Materials: Written materials must be adapted to suit the literacy level of implementation agents and caregivers. Materials should be adapted to accommodate and reflect community norms and language. The use of local songs, proverbs, and picture books can be appropriate for children and caregivers. Homemade materials can serve as toys. Written guides for implementation agents promote program quality and workforce acceptance. | (30) | |
| | | IS3.6 | Caregiver Engagement: Intervention modules, session delivery site and timing, and recruitment strategies can be adapted to include diverse caregivers, including grandparents and fathers as well as mothers. Communication about the intervention must emphasize the program's ECD goals and encourage caregiver participation. High intervention quality that leads to noticeable improvements in child development can drive caregiver acceptance of and participation in the program in addition to trust of implementation agents on a range of issues. Caregivers were enthusiastic about the intervention when seeing ECD improvements in their young children and sometimes also the older siblings. Programs should take into consideration perceptions of equity within the family when providing a program to just one sibling. | (15) | |
| | | IS3.7 | Supplementary Program Activities: Intervention content could integrate child health and development interests of the community and consider existing caregiving practices and challenges (e.g., breastfeeding, nutrition, maternal psychosocial wellbeing). Intervention modules can integrate other domains of nurturing care but must take care to not be too unwieldy or unfeasible. | (21) | |
| | | IS3.8 | Responsive Teaching: Implementation agents can actively engage caregivers in responsive activities during sessions to facilitate and encourage continued practice between and following sessions. | (15) | |

(n = 17), and when necessary rapid modifications or adaptations (n = 2). A monitoring system was found to be critical to scaling (n = 3), and sustainability (n = 6).

## Pathways to scale up the Care for Child Development (CCD) and Reach Up (RU) Programs

Fig 3 displays the refined evidence-based program impact pathways analysis that illustrates the relationships among context, implementation strategies, and outcomes across the six identified building blocks for successful program implementation and scale up.

*Program initiation* includes the emergence and intersectoriality building blocks related to country readiness. It relies on securing sustainable funding sources, conducting a landscape analysis of the existing ECD system for the potential of service integration, and putting in place strong intersectoral leadership and governance structures for multi-level program implementation. Next, *program design* includes elements of the intervention characteristic and system monitoring building blocks. In this phase, key design decisions must be made, including how the CCD or RU responsive caregiving curriculums can be integrated with services

**Table 11. Intervention characteristics: Syria, Lebanon, and Jordan (Reach Up).**

In 2017, the International Rescue Committee began delivering an adapted RU curriculum to displaced Syrian families with children under the age of three living in Syria, Lebanon, and Jordan. While program stakeholders and implementation approaches varied across the three country sites, the program materials were consistent. The conflict-affected setting including displaced populations posed unique, significant challenges to program delivery; including damaged road infrastructure, shortages of material, political instability, and widespread trauma experienced by both home visitors and participating families. Each country team was composed of home visitors and supervisors, and intervention dose and frequency ranged from 30-minute fortnightly sessions in Jordan to 60-minute weekly sessions in Syria. This flexibility allowed local implementers to choose the most feasible delivery characteristics for their setting. All visits followed the general steps of greeting, catching up, reviewing the previous session's activities, introducing new activities, singing a song, and a final review of the session's material. The RU curriculum used by home visitors was professionally translated into Arabic and reviewed by the Arabic Resource Collective before additional adaptations were made by country teams to maximize local relevance. Home visits included the use of child-safe toys made from locally available material, such as dolls made of socks, and alternative toys and materials described for the home visitor to use if the first-choice materials were not available. Children's books were produced with images of tent-style living to reflect the homes of refugee children and with little or no text to meet the appropriate literacy level of the population. Additional materials were created to be used for families with children with disabilities. The team delivering the program in Syria was traumatized by the sustained conflict and dynamically responded to the ongoing emergency by prioritizing the supportive relationship between the home visitors and families to build trust and more effectively introduce responsive caregiving activities when families were able to receive services. The systematic adaptation of program materials and flexible program delivery mechanism of the RU program in these conflict-affected countries were essential to program adoption, appropriateness, acceptance by families and implementers, and general feasibility.

addressing other nurturing care domains based on the unique ***program context***. Lastly, ***program delivery*** cycles through stages of preparation, sustainment, and assessment. During program delivery, the workforce, training, monitoring systems, and intersectoriality building blocks actively and jointly determine implementation outcomes and scaling impact (Fig 3).

The ECD Implementation Checklist for Enabling Program Scale up (Table 18) (hereafter ECD Program Implementation Checklist), that we developed as part of this study, evolved out of the revised evidence-based program impact theory, and presents a set of questions, organized by building blocks, to guide informed decision-making on program implementation strategies that in turn can lead to the enhanced implementation outcomes.

## Discussion

Our scoping review documented the context, implementation strategies, and implementation outcomes shaping implementation impact pathways of Reach Up (RU) and Care for Child Development (CCD) in LMICs. We identified six interrelated building blocks representing the components needed to successfully implement ECD nurturing care programs as multisectoral integrated interventions on a large scale [21, 36, 102]. With the construction of the evidence-based program theory and development of the ECD Program Implementation Checklist, our findings address the major gap in evidence on best practices for strategic planning, adapting, delivering, and scaling effective and sustainable ECD programs. The revised evidence-based program impact theory should be used and interpreted as an iterative process, with program initiation and co-design decisions revisited often throughout the program delivery phase to adapt the program to emerging context challenges such as baseline inequities in early life, especially among the most vulnerable populations such as low-income young families [103] and single mothers [104]. The ECD Program Implementation Checklist provides a user-friendly dynamic, non-exhaustive tool for implementers to effectively design, implement, scale up and sustain an ECD nurturing care program. This Checklist can help Implementers identify challenges on each building block, noting any affirmative or negative responses to suggest whether

**Table 12. The workforce building block of Care for Child Development (CCD) and Reach Up (RU) implementation pathways.**

| Context | | Implementation Strategies (Number of programs reporting) | | | Implementation Outcomes (number of programs reporting) | |
|---|---|---|---|---|---|---|
| **Facilitators** | **Barriers** | | | | | |
| ECD System (Workforce): The presence of a pre-existing workforce and/or health services infrastructure affected program delivery in some cases. | ECD System (Workforce): An overburdened and/or overworked workforce was not as receptive to delivering a CCD intervention | IS4.1 | Workforce Recruitment: A new cadre of workers or providers trained in other ECD services (e.g., educators, community health workers) can be trained to deliver the program. The use of an existing workforce to engage and hire implementation agents is typically feasible and cost-effective. Training a new cadre of workers can also be feasible and cost-effective with sufficient support, and training, and when using community-based workers instead of highly educated interventionists. Often implementation agents who were female, and local community members, with at least basic education and literacy were highly acceptable. | (32) | Appropriateness | (6) |
| | | | | | Feasibility | (13) |
| | | | | | Acceptability | (13) |
| | | | | | Adoption | (7) |
| | | | | | Fidelity | (7) |
| | | | | | Adaptation | (0) |
| | | | | | Penetration | (1) |
| | | | | | Sustainability | (6) |
| | | | | | Implementation Cost | (6) |
| | | | | | Scaling | (3) |
| Infrastructure: Implementation agent activities were impacted by logistical challenges such as poor roads, electricity outages, heat, and local conflict or violence. | Workforce Motivation: If implementation agents did not feel supportive, valued, or effective they were less motivated to deliver high-quality CCD or RU sessions. | IS4.2 | Implementation Agent Training: Implementation agents can be trained to be strategic and thoughtful about the engagement of key decision-makers in the household; they should also feel confident in conducting visits with mothers and caregivers who are potentially more educated or have a higher SES. Initial training to establish skills is critical to successful program delivery. Training should include practicing empathetic listening and caregiver encouragement. Culturally specific and organized manuals and curriculum materials can be used for training and setting implementation agent expectations regarding their role. | (32) | | |
| Hiring Pool: Qualifications and education level of implementation agents varied by area and context. | | IS4.3 | Job Satisfaction: Implementation agents who felt prepared and supported by supervisors, had access to easy-to-use tools and guidelines, and observed improvements in the children and/or caregivers they worked with felt useful and that they had ownership over the program. Implementation agents hired from an existing intersectoral workforce were motivated to deliver the program when they recognized how the program supported their pre-existing tasks. | (11) | | |
| Social Norms: Community dynamics, customs, and gender roles influenced how implementation agents interacted with and were received by the community. | | IS4.4 | Compensation and Contracts: The compensation of implementation agents should be comparable to similar workforce compensations in a given setting. Providing adequate compensation or stipends for program delivery was essential to workforce satisfaction and high-quality program delivery. Contracts must be of a sufficient length of time to ensure workforce stability and satisfaction. | (16) | | |
| | | IS4.5 | Caseload: A manageable caseload, influenced by existing responsibilities of the workforce and the geography of the catchment area, is essential to program quality and fidelity. | (13) | | |
| | | IS4.6 | Workforce Characteristics: Female implementation agents were typically acceptable to female caregivers while male implementation agents were sometimes respected, but it was occasionally seen as inappropriate for mothers to interact with them alone. Implementation agents from a lower social class or with less education than mothers had lower confidence in their role. Programs can be delivered by lay community members or more highly educated workers depending on the context. | (21) | | |
| | | IS4.7 | Community integration: Having a workforce be familiar with and accepted by the community-made caregivers more comfortable with their presence and the intervention overall. The visibility of the workforce and the program's positive outcomes drove community trust in both the workforce and intervention. | (20) | | |

**Table 13. Workforce: Malawi (Care for Child Development).**

Malawian researchers and international organizations partnered with government ministries to design and implement a locally adapted CCD program beginning in 2013. After giving careful consideration of the appropriate workforce for large scale program delivery, the program implementers carried out a formal job analysis of community health workers from different sectors and a formative qualitative assessment that optimized the CCD material for the local setting. A feasibility trial of a program using the CCD curriculum was then conducted using community health workers, called Health Surveillance Assistants, to deliver the intervention to families. Health Surveillance Assistants were respected members of their community, trained by the Ministry of Health and provided with a moderate government stipend to conduct various community-based health promotion activities supporting family health, environmental health, child nutrition, and control of communicable diseases. In addition to a national policy environment supportive of integrated nutrition and ECD interventions, the appropriateness and availability of the workforce to deliver this new ECD intervention was dependent on program bureaucratic factors, strategic task allocation, and coordination of work schedules. The feasibility trial revealed that Health Surveillance Assistants could not feasibly deliver the integrated responsive caregiving and nutrition intervention, given their unrealistic caseload due to pre-existing work activities, and low compensation. While the use of group sessions allowed for Health Surveillance Assistants to effectively use their limited time by delivering curriculum to many participating families, the program still had low penetration. Each of the Health Surveillance Assistants was only able to deliver the integrated program to a caseload of ten families, which translated into less than 15%of the eligible children in their catchment areas. While the program activities were acceptable to participating families and Health Surveillance Assistants, they were not feasible to deliver using this pre-existing workforce. Investigators recommended a different delivery model using Health Surveillance Assistance in a supervisory capacity, with the program delivery implemented by a different cadre of community-based workers that were already being employed by the Ministry of Gender and Social Welfare.

or how their program has accounted for these enabling mechanisms. Importantly, this guide is designed to be used together with context specific implementation pathways to maximize the chances of success scaling up the ECD program of interest.

In this study we documented variations in how governments pilot, test the feasibility, adapt, operationalize, and scale up integrated ECD nurturing care programs in their specific contexts. Consistent with prior evidence, we found that implementation decisions rely on understanding how to coordinate existing services, political will, geography, and available human and financial resources since program conceptualization [105]. Therefore, programs varied greatly in aims, scope, and impact. Accordingly, key contextual factors inform the design and development of context-sensitive interventions, and recognition of this influence is critical to effective, sustainable, and scalable implementation [106]. By recognizing how contextual factors generate complex and dynamic interaction among sectors or systems, we acknowledge that a program's context is not fully controllable or predictable but at the same time that using implementation science principles can properly inform how the intervention may be continually targeted, and effectively adapted alongside evolving circumstances [37, 106].

The Nurturing Care Framework includes the five mutually supported ECD domains of health, nutrition, early learning, safety and security, and responsive caregiving that need to be integrated for effective scale up of ECD services [9, 35]. Our findings demonstrate the pathways through which CCD and RU programs can place responsive caregiving at the center of behavior change around which the other nurturing care domains may be scaffolded and synchronously addressed. While CCD and RU are both included in this review due to their parallel goals and similar approaches for promoting responsive caregiving, they differ slightly in their origins, curricular content, evidence-base, and real-world utilization. Compared to RU, CCD has been more likely to be implemented on a larger scale due to its promotion by the WHO [10, 20]. By contrast, RU is backed up by relatively more robust feasibility and randomized controlled studies spanning decades [37, 80, 84, 90, 92–94, 96, 101]. An example of this can be seen in Brazil, as far as we know the only country included in this review that has real world experiences implementing both RU and CCD [20, 80]. Brazil's national Criança Feliz

**Table 14. Training building blocks of Care for Child Development (CCD) and Reach Up (RU) implementation pathways.**

| Context | | Implementation Strategies (Number of programs reporting) | | | Implementation Outcomes (number of programs reporting) | |
|---|---|---|---|---|---|---|
| **Facilitators** | **Barriers** | | | | | |
| ECD System (Infrastructure): Existing workforce development activities (e.g., IMCI training of clinicians) and program training practices can be used as a platform to integrate program training. | External Pressure to Scale: the challenge of maintaining training cascade capacity and sustainability when a program is rushed to scale. | IS5.1 | Training Manuals and Protocols: Having a standardized, evidence-based program curriculum using different materials and modalities (e.g., role-play, films) was critical to successful program training. As needed, training can be adapted to include additional activities such as nutrition counseling or maternal mental health; however, the inclusion of such additional training was not consistently reported, and the overall extent and nature of these adaptations are unclear. | (18) | Appropriateness | (12) |
| | | | | | Feasibility | (11) |
| | | | | | Acceptability | (11) |
| | | | | | Adoption | (3) |
| | | | | | Fidelity | (5) |
| | | | | | Adaptation | (7) |
| | | | | | Penetration | (0) |
| | | | | | Sustainability | (9) |
| | | | | | Implementation Cost | (3) |
| | | | | | Scaling | (5) |
| Environment: Due to the changing social and material environment of intervention delivery (e.g., electricity outages, interruptions by family members, local political conflict, etc.), the training must emphasize the need for flexibility. | Training Leadership: Programs were less sustainable and scalable in cases where training was led by external experts instead of local leaders. | IS5.2 | Training Skills: Training must build implementation agents' skills and flexibility so they can adapt to various potential circumstances they may encounter while delivering the program. Training must include behavior change methods, empathetic listening, flexible problem solving, how to clearly communicate program goals, and how to answer caregiver questions. | (15) | | |
| Availability: The time and resource availability of implementation agents and supervisors must be considered. | | IS5.3 | Continuing Education: Ongoing training during program delivery can provide opportunities to introduce new content and scaffold upon existing skills (e.g., responsive caregiving). This can support programs being delivered with fidelity and the ability to provide any needed program improvements. | (16) | | |
| | | IS5.4 | Training Leadership: An effective training cascade consists of master trainers internally hired and trained. Supervisors sometimes led implementation agent training in addition to their ongoing supervisory tasks. | (14) | | |
| | | IS5.5 | Training Design: In-person training was typical, but virtual training was successful at times and allowed more implementation agents to be trained. Training length varied by site (from 2 days to several weeks). | (26) | | |
| | | IS5.6 | Institutionalization: Training can be embedded in existing ECD services and workforce development programs if the program itself is institutionalized. | (7) | | |

program, which is based on CCD, has been rapidly scaled up as a result of strong political will and national leadership. However, our analysis shows that there have been strong challenges for intersectoral collaboration, including the challenges of developing a sustainable referral network to link the program with existing social services [20]. Interestingly, the RU program implemented in the city of São Paulo (Brazil) operated on a much smaller scale and successfully used an intersectoral workforce of community health agents to deliver the RU curriculum [80]. While this was feasible on a small scale, the experience of Brazil's Criança Feliz illustrates the challenges of intersectoral collaboration on a national level which is consistent with a previous study from our group [20]. This example demonstrates that the implementation strategies used to deliver CCD or RU were stronger determinants of successful implementation outcomes than the program itself [34, 107]. Therefore, countries can benefit from the knowledge generated in this study about the CCD and RU key to help them develop a comprehensive and effective nurturing care system.

**Table 15. Training: Peru, Cuna Más (Reach Up).**

In 2012, the Peruvian Ministry of Development and Social Inclusion collaborated with various government ministries to form the national ECD program, Cuna Más, implemented nationally in rural areas with high levels of child stunting and poverty, measured by prevalence of conditional cash transfer recipient families. As of 2016, the program reached 32% of eligible participants totaling 85,000 families. The program employs and trains cadres of local community members as home visitors to provide individual home visits and group sessions using the RU methodology. Program operations are supported and coordinated by local management committees made up of nominated community members. A training cascade model is used wherein central program staff and specialists train and supervise regional staff, who are in turn responsible for the training and supervision of local technical companions and home visitors. Training of home visitors is led by trainers with support from regional ECD specialists and technical companions who serve as immediate supervisors of the home visitors. Regional trainers lead a nine-day pre-service training for new home visitors, with additional in-service training provided once or twice a year. Additionally, there is ongoing technical assistance available to home visitors. Manuals were developed to guide the training and delivery of program activities including home visiting guides for different levels of staff, ECD surveillance protocols, enrollment and graduation of families, in addition to an overarching operational manual. The cascade training model allows for trainings to be tailored to the local setting by incorporating input from different program level staff to improve appropriateness and acceptability. One barrier to implementation was the confusion of program staff regarding poorly defined work roles and responsibilities that were not clearly established during training. This challenge, in combination with unsatisfactorily temporary contracts, contributed to the high turnover rate of 33% of home visitors in 2016. Home visitors and other staff valued the training and ongoing support from supervisors, which was seen as essential to workforce program acceptability, intervention fidelity, and staff cohesion.

Through the process of documenting the building blocks, we found that complex implementation details were inconsistently reported, and sometimes they were not even reported. For example, almost all programs in our review provided details regarding intervention characteristics likely to be more easily documented and measured, such as the setting and delivery, materials, and dose and frequency. By contrast, while most authors acknowledged the importance of reporting implementation strategies and implementation outcomes, there was an almost complete lack of specific implementation details reported, especially in relationship to the intersectoriality, program emergence and monitoring building blocks. This indicates that the use of frameworks and standardized tools to better report implementation studies and programs is critical to further advance knowledge in the field of ECD programming based on robust implementation science principles [17, 108].

Our review documented consistent evidence informing features of program emergence for both RU and CCD implementation sites. Funding was the feature most often reported as a barrier to sustainability, which is consistent with previous studies [8, 20, 45]. On the other hand, there was less information regarding how the scaling process and governance and leadership were considered at each stage of implementation. Implementation studies focusing on Criança Feliz and Chile Crece Contigo have demonstrated the importance of the governance system for successful scale-up of integrated ECD nurturing care programs [20, 109]. This is especially true for programs designed to benefit vulnerable populations facing a constellation of early life adverse experiences, necessitating well- coordinated action from numerous sectors simultaneously. Existing literature substantiates our findings that while on the one hand most programs claim the importance of intersectoral collaborations [26, 54, 76, 85], on the other hand very few studies have examined if the programs were conceived and implemented through an intersectoral lens. Therefore, moving forward our findings call for a much more systematic analysis of multisectoral governance and coordination of implementation systems at different stages of the scaling processes of ECD programs [36].

Almost all programs in our review reported details regarding workforce development. Consistent with previous studies, high personnel turnover, low salaries, and short-term or temporary contracts have been identified as major challenges for the proper implementation

**Table 16. Monitoring building blocks of Care for Child Development (CCD) and Reach Up (RU) implementation pathways.**

| Context | | Implementation Strategies (Number of programs reporting) | | | Implementation Outcomes (number of programs reporting) | |
|---|---|---|---|---|---|---|
| **Facilitators** | **Barriers** | | | | | |
| ECD System (Supervisor Workforce): Supervisors can employees recruited from other ECD programs. If the program is delivered by an existing workforce, that workforce's existing supervisory staff may be used. | Infrastructure: Logistical challenges result when supervisors are responsible for implementation agents across a wide geographic area and cannot easily conduct site visits. | IS6.1 | Supportive Mentoring: Supportive mentoring and support of implementation agents (i.e., praise, encouragement) by supervisors during site visits, regular in-person meetings, telephone calls, etc., are critical to satisfaction, intervention fidelity and quality, and implementation agent engagement. Supervisors often used fidelity checklist tools to tailor their guidance for individual implementers based on weak areas. This helped guarantee program quality and expand the implementation agent's skill set. Supervisors can inform and participate in the improvement of the formal implementation agent training to address performance weaknesses, challenges, or gaps in training that they observe in the field and support the success of future implementation agents. | (25) | Appropriateness | (5) |
| | | | | | Feasibility | (7) |
| | | | | | Acceptability | (1) |
| | | | | | Adoption | (1) |
| | | | | | Fidelity | (17) |
| | | | | | Adaptation | (2) |
| | | | | | Penetration | (0) |
| | | | | | Sustainability | (6) |
| | | | | | Implementation Cost | (0) |
| | | | | | Scaling | (3) |
| Program Scale: Smaller pilots had more informal, less developed supervision structures, whereas larger-scale implementation requires more advanced supervisory infrastructure. | | IS6.2 | Standardized Data Collection Tools: Tools such as attendance sheets, data collection forms, and audiotapes of sessions were used by supervisors and implementation agents to monitor intervention delivery, caregiver participation, and ECD outcomes. These tools must be designed to facilitate detailed data collection while not over-burdening implementation agents and supervisors. Electronic monitoring systems can be used to store program data and provide easy access to data to program leaders to inform future program decisions. | (31) | | |
| | | IS6.3 | Data-Informed Decision Making: Program data are organized and managed by senior leadership to inform stakeholders and make evidence-based programmatic decisions. | (20) | | |
| | | IS6.4 | Supervisor Training: Supervisors should be trained intensively on the program as well as their supervisory role of collecting and managing data, supporting implementation agents, and disseminating program information as needed. | (18) | | |
| | | IS6.5 | Supervisor Caseload: Supervisors who are responsible for too many implementation agents or too many tasks (i.e., too large a caseload) are ineffective. A manageable caseload was influenced by the existing responsibilities of the supervisor and the geography of their catchment area. | (11) | | |
| | | IS6.6 | Impact Measurement: The impact of the program on ECD can be measured by integrating child development indicators into existing national health surveillance systems, such as data collected by public health facilities. | (30) | | |

of ECD programs [18, 20, 82]. Our study identified additional barriers related to compensation and contracts, job satisfaction, and caseloads, which fills in an important gap as there has been a dearth of studies documenting factors related to the success or failure of implementation strategies related to workforce training, retention and motivation [18, 82]. The training program design, including curriculum and cascade, was the most common feature reported. However, specifications regarding manuals, protocols, training leadership, and adaptations to local contexts were scarce. Likewise, none of the studies included in our review assessed the impact of staff training on the expected skills and competencies, which is considered critical for both CCD and RU program implementation fidelity and quality [20].

**Table 17. Monitoring systems: India, Project Grow Smart (Care for Child Development).**

Project Grow Smart is an ECD program integrating the CCD methodology with micronutrient supplementation implemented in rural areas of India from 2012 to 2013. The program was developed following formative research in which local women indicated that their goals were for their children to grow up smart and healthy which then inspired the name of the program. The program used newly trained village women to deliver the intervention to local families with children between the ages of six and twelve months. Project Grow Smart had a robust supervisory and monitoring system supporting program activities. Four supervisors with university degrees and backgrounds in child development were recruited and trained to oversee approximately nine implementation agents. Supervisor training lasting one month was led by the study investigators and used various teaching techniques including role playing to cover program operating procedures, the use of monitoring forms, and techniques for supporting implementation agents. Supervisors met with individual implementation agents fortnightly to help problem solve, replenish program supplies, and review progress of participating families. On a monthly basis, supervisors observed implementation agent sessions with families and completed an observation record checklist which informed tailored feedback and program records. The implementation agents also completed regular home visit evaluation forms to document program activities and family progress. Supervisors reviewed these forms to measure fidelity and address implementation agent challenges and weaknesses. The strong program supervision led the implementation agents to have high acceptability of their role due to the support they received. The strong relationships between supervisors and implementation agents in combination with systematic monitoring mechanisms improved the quality of the intervention and ability of program leaders to make data-driven decisions.

Corroborating these findings, a recent implementation study found that an upfront investment in training local trainers and delivery agents as well as regular manualized supervision were critical to the successful implementation of an ECD program in rural Kenya [43]. Most programs mentioned supportive supervision, supervisor training, standardized data collection, and impact measurement tools as important for monitoring the implementation of CCD and RU. However, several elements central to a monitoring system, including supervisor caseload and training, were not specified in the available literature. A recent global literature review identified similar problems with implementation fidelity and quality of CCD as a result of difficulties with the training of the workforce on this home visiting program [22]. The poor quality of ECD program implementation represents a critical and major barrier negatively affecting long-term program sustainability in LMICs [21]. Our scoping review and innovative implementation science analysis builds on and substantially adds to this

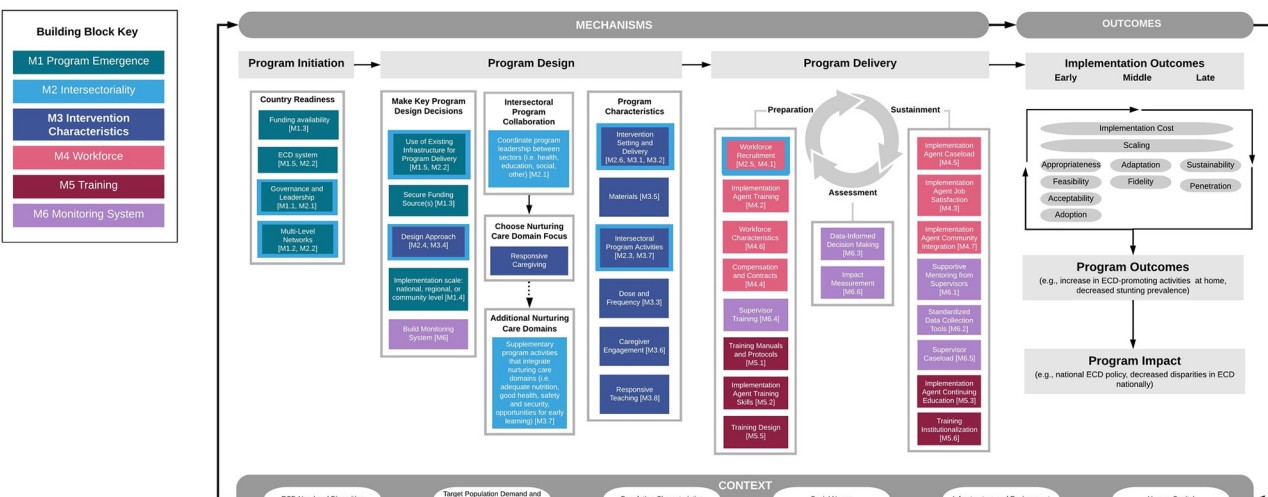

**Fig 3. Refined evidence-based program impact theory for successful implementation and scale up of Early Childhood Development Programs.**

**Table 18. ECD implementation checklist for enabling effective program scale up.**

| Implementation Strategies (IS) | | Building Block (BB) |
|---|---|---|
| **BB1 Program Emergence** | | |
| IS1.1 | Governance and Leadership | **Are there individuals and/or groups to advocate for the program on a national level?** Have individuals, groups, or institutions that will coordinate program implementation been identified? Is ECD programming supported by ECD legislation and policy? What government sector or non-government agent has ownership over this program and its implementation? |
| IS1.2 | Multi-Level Networks | **Are there partnerships for implementation, coordination, and program planning at multiple levels (i.e., national, regional, local)?** How can these partnerships establish methods of communication and collaboration? |
| IS1.3 | Funding Availability | **Is there a sustainable funding stream for program implementation?** Is the funding from internal (i.e., local government) or external sources (i.e., international donors, research funding)? |
| IS1.4 | Scaling Process | **Is there an intentional process in place for rolling out the program?** Has a pilot or feasibility study been conducted? How will program quality be maintained through the scaling process? |
| IS1.5 | Integration into ECD System | **Is there a way for the program to be embedded in existing government or non-governmental infrastructure(s) that serve children and families?** Are there existing services (i.e., healthcare, early education, conditional cash transfer programs) supporting the needs of families? Is there an existing workforce that is open to delivering additional ECD programs? |
| **BB2 Intersectoriality** | | |
| IS2.1 | Coordination and Communication | **Do intersectoral stakeholders have clearly designated commitments and responsibilities?** How will intersectoral stakeholders formally and informally regularly communicate with one another? How will messaging about the program be consistent from all program partners? How will local adaptations be designed to match local capacity and context? How will program implementers refer families to other ECD service providers to meet additional family needs? |
| IS2.3 | Intervention Characteristics | **Could the program be delivered along with other interventions for child health and development (e.g., nutritional supplementation)?** How might these interventions be most effectively integrated? |
| IS2.4 | Intersectoral Targeting | **Is there an existing mechanism for classifying vulnerable families (e.g., conditional cash transfer eligibility) that could be used to selectively target CCD or RU recipients?** How might such a system be initiated or operationalized to support an ECD program? |
| IS2.5 | Intersectoral Workforce | **Is there an existing network of accessible facilities to target children and families that could be used as the delivery sites of CCD or RU?** How could program tasks be feasibly incorporated into existing staff roles and responsibilities? |
| IS2.6 | Delivery Sites and Networks | **Is there an established health system network at the community level?** Are there existing facilities that are familiar to target families that could be used for program delivery? Are there other acceptable, accessible sites for delivery (i.e., schools, community centers, churches)? |
| **BB3 Program Characteristics** | | |
| IS3.1 | Intervention Delivery | **Will the program be delivered in a group size that is feasible and acceptable to program participants and implementers?** Will the program be delivered via sessions with individual families, groups of families, or a combination of individual and group sessions? How will the mode of delivery create a space in which caregivers are comfortable interacting, sharing, and asking questions? |
| IS3.2 | Intervention Setting | **Has an appropriate setting been determined to deliver the intervention?** Will the program be delivered in families' homes and/or in another local setting, and why? Is the intervention setting accessible by locally available transportation and appropriate for the families? |
| IS3.3 | Dose and Frequency | **Has a plan been designed to provide sustained delivery of sessions?** Is there a need for booster sessions? What program adaptations are necessary to determine and/or achieve the intervention dose and frequency? |
| IS3.4 | Design Approach (universal, selective, indicated) | **Will the program be delivered universally to all families in a catchment area, selectively to a certain classification of families, or only to children who are screened and are indicated as eligible?** Which children/families should be prioritized to receive the program? Are children and families targeted through the use of an existing mechanism (e.g. conditional cash transfer)? How is the intervention targeting process going to be transparent and equitable? How are eligible children/families being reached as early as possible? |

*(Continued)*

**Table 18.** (Continued)

| Implementation Strategies (IS) | | Building Block (BB) |
|---|---|---|
| IS3.5 | Materials | **Were materials adapted to the program context (i.e., considerations of language, literacy level, resources, cultural appropriateness)?**<br>How do materials reflect community norms?<br>What materials are locally available? |
| IS3.6 | Caregiver Engagement | **Do program modules promote engagement of fathers and other caregivers in addition to mothers?**<br>How are different caregivers supported to engage with the curriculum?<br>How can the program delivery be adaptable to individual family context and need? |
| IS3.7 | Supplementary Program Activities | **Will the core curriculum program content be integrated with additional content on other nurturing care domains (e.g., nutrition counseling)?**<br>Is there a demonstrated need to address other nurturing care domains?<br>What activities or programs already exist that could feasibly be integrated with the program, and how? |
| IS3.8 | Responsive Teaching | **Will program content be delivered to families in an interactive, engaging way by the implementation agents?**<br>What relevant methods, approaches, and techniques do implementation agents need to be trained in to effectively deliver the content? |
| **IS4 Workforce** | | |
| IS4.1 | Workforce Recruitment | **What group of individuals will be trained as implementation agents to deliver RU or CCD?**<br>Is there an existing, accessible workforce that can integrate the intervention into their existing activities (e.g., clinicians, community health workers, early childhood educators)?<br>How will potential implementation agents be identified? |
| IS4.2 | Implementation Agent Training | **Does the training prepare implementation agents to:**<br>(a) Adapt and deliver the program as needed to reflect family needs<br>(b) Work with potentially complicated households and family dynamics<br>(c) Build rapport with family members<br>(d) Practice empathetic listening and effective communication |
| IS4.3 | Job Satisfaction | **Is the implementation agents' satisfaction prioritized when making other program decisions (i.e., determining caseload, training, program material, and supervision)?**<br>What motivates implementation agents and how can they be best supported by the program? |
| IS4.4 | Compensation and Contracts | **Are implementation agents appropriately compensated?**<br>What forms of compensation are most appropriate and desirable in a given setting?<br>Is the corresponding contract length adequate and appropriate to sustain the workforce? |
| IS4.5 | Caseload | **Is there a determined number of program-recipient families that each implementation agent will be responsible for?**<br>Is this caseload feasible given potential logistical challenges and travel?<br>How can the caseload be flexible to accommodate implementation agents' existing commitments? |
| IS4.6 | Workforce Characteristics | **Are there specific implementation agent characteristics (e.g., gender, age, education) that make them more or less effective at program delivery in a given setting?**<br>How might certain characteristics such as gender, social class, or education level shape their ability to effectively perform their job? |
| IS4.7 | Community Integration | **Are implementation agents already familiar with the area and respected members of the community they will be working in?**<br>What actions can be taken to build and strengthen the community trust of the implementation agents? |
| **BB5 Training** | | |
| IS5.1 | Training Protocols and Manuals | **Are there manuals or protocols to guide the implementation agent training?**<br>How are the training protocols and manuals adequately adapted to fit the specific setting and workforce needs?<br>Are the training protocols and manuals consistent and evidence-based? |
| IS5.2 | Training Skills | **Is there sufficient emphasis on training implementation agents in the skills necessary to be effective in their role?**<br>How will the training build implementation agents' skills in behavior change, empathetic listening, problem-solving, and communication that enable effective delivery of CCD or RU?<br>How will the training build implementation agents' skills to be flexible and adapt to emerging needs and challenges that arise during program delivery? |
| IS5.3 | Continuing Education | **Are there opportunities for continuing education for the workforce?**<br>Following initial pre-service training of implementation agents, what additional training opportunities can be provided to address emerging needs or transfer additional information? |
| IS5.4 | Training Leadership | **Is there an established system to deliver the training to the implementation agents?**<br>Will there be a training cascade in which master trainers are equipped to deliver the training to implementation agents?<br>What percentage of the training and leadership team are internal or external to the community being served? |

*(Continued)*

**Table 18.** (Continued)

| | Implementation Strategies (IS) | Building Block (BB) |
|---|---|---|
| IS5.5 | Training Design | **Is there a set training design that supports the implementation agents' needs?** What length of training is feasible and sufficient to prepare implementation agents to deliver the program? Will the training be held in a central location, various locations across the program delivery setting, or conducted virtually? |
| IS5.6 | Institutionalization | **Is the training embedded into existing workforce development programs within the ECD sector of the program delivery setting?** How will the training be institutionalized to support program sustainability and scaling? |
| | **BB6 Monitoring Systems** | |
| IS6.1 | Supportive Mentoring from Supervisors | **Is supportive supervision built into the program?** What techniques do supervisors use to provide ongoing encouragement and support to implementation agents? How often are supervisors and implementation agents meeting to hold one-on-one feedback sessions? How can supportive mentoring help improve program fidelity, quality, and effectiveness? If supportive mentorship is provided, how will supervisors uniquely tailor feedback and guidance for implementation agents to address specific needs and areas of improvement? |
| IS6.2 | Standardized Data Collection Tools | **Are there tools that the implementation agents and supervisors will use to collect data on program implementation and outcomes?** When and how will supervisors and implementation agents be trained to use the appropriate tools? How will these tools be flexible and adaptable? |
| IS6.3 | Data-Informed Decision Making | **Is there a designated party responsible for program data collection management?** How will data be used to inform decisions about the program? To whom else does the data need to be shared? |
| IS6.4 | Supervisor Training | **Will supervisors be trained to provide high-quality supervision?** How will this training be integrated into the program training scheme? Who will provide the training? How will the training be evaluated? |
| IS6.5 | Supervisor Caseload | **Has a caseload ben determined?** How many implementation agents can each supervisor feasibly support given the setting context? How are supervisors held accountable for supporting their caseload? |
| IS6.6 | Impact Measurement | **Has it been determined which impacts will be measured?** How will the program impact outcomes (i.e., child health, child development, family stability, etc.) be measured among families enrolled in the program? |

prior evidence as it provides new critical insights into best practices implementation strategies for workforce training, supportive supervision, and program monitoring during CCD and RU in low- and middle- income countries.

Although our review has filled in important knowledge gaps in the field of integrated ECD programming in LMICs, these findings should be interpreted with caution. First, our evidence synthesis was limited by the publicly available data about each program, although we designed our scoping review to collect as much evidence as possible to understand how to successfully implement ECD interventions across LMIC settings. Second, the scope of this review is limited to RU and CCD, the programs with the strongest evidence base and the highest coverage across LMICs. As a result, programs based on other responsive caregiving curriculums such as Learning Through Play, Triple P Positive Parenting Program, Parenting our Children to Excellence, Let's Talk about Children, and Attachment and Behavioral Catch-up were beyond the purview of this review. Our review used an innovative thematic analysis approach to synthesize diverse forms of evidence to generate a widely applicable, evidence-based program impact theory and a Checklist can be used by ECD program implementers in LMICs in conjunction with other available resources and tools to support the scaling-up process. While the goal of this review was to identify implementation pathways contributing to successful scale up, we collected evidence on programs at all levels of scale to identify common barriers and adaptations to local

contexts. Future reviews should further investigate implementation pathways for other multi-sectoral responsive caregiving and parenting skills programs in LMICs, especially those implemented at scale including the national level. To support these reviews, implementation evaluations of each site should be rigorous, with consistent data collection and reporting. It is crucial that implementers and researchers proactively document the implementation of the strategies embedded in their ECD interventions to illuminate how implementation outcomes are reached in diverse countries. Therefore, we recommend implementers to use the ECD Program Implementation Checklist that we developed as part of this study. Furthermore, we recommend that ECD program implementers consistently report implementation approaches and evidence to facilitate efficient sharing of knowledge with fellow implementers. The Consolidated Advice on Reporting ECD implementation research (CARE) is a systematic approach to reporting of implementation processes, theory of change, fidelity, and adaptation of nurturing care interventions for promoting early child development [110]. Other systematic approaches non-specific to ECD can be adapted to report program details such as the Template for Intervention Description and Replication (TIDieR) [17] and the Standards for Reporting Implementation Studies (StaRI) [108]. Moving forward, it is necessary to understand how integrated ECD programs influence nurturing care environments across households, communities, and countries to best support children's health and wellbeing [111–113].

## Conclusions

We identified implementation pathways organized into six building blocks for the successful implementation of CCD and RU nurturing care programs in LMICs. Our review calls for the careful consideration of integrated ECD program leaders to the importance of the context to develop implementation strategies that enable successful implementation and, ultimately, reduce early adverse experiences and provide stimulating nurturing opportunities since gestation. Implementers can use the ECD Program Implementation Checklist that resulted from our study to strengthen their program's intersectoral nurturing care implementation strategies and outcomes.

## Supporting information

**S1 Checklist. Preferred Reporting Items for Systematic reviews and Meta-Analyses extension for Scoping Reviews (PRISMA-ScR) checklist.**
(DOCX)

**S1 Fig. Initial program theory.**
(TIFF)

**S1 Table. Description of RU and CCD methodologies using the Template for Intervention Description and Replication (TIDieR).**
(DOCX)

**S2 Table. Search terms.**
(PDF)

**S3 Table. Data extraction items.**
(DOCX)

**S4 Table. Qualitative data collection of implementation strategies of Care for Child Development programs.**
(PDF)

**S5 Table. Qualitative data collection of implementation strategies of Reach Up programs.**
(PDF)

**S6 Table. Qualitative synthesis of implementation outcomes of Care for Child Development programs.**
(PDF)

**S7 Table. Qualitative synthesis of implementation outcomes of Reach Up programs.**
(PDF)

**S8 Table. Implementation strategies screening by building block.**
(PDF)

**S9 Table. Implementation outcome screening by building block.**
(PDF)

**S10 Table. List of included records for Care for Child Development and Reach Up programs.**
(DOCX)

## Acknowledgments

The authors would like to acknowledge the very valuable contribution of the Yale School of Public Health Librarian Kate Nyhan in the conceptualization and implementation of the review search strategy.

## Author Contributions

**Conceptualization:** Gabriela Buccini, Rafael Pérez-Escamilla.

**Data curation:** Gabriela Buccini, Lily Kofke, Haley Case, Marina Katague, Maria Fernanda Pacheco.

**Formal analysis:** Gabriela Buccini, Lily Kofke, Haley Case, Marina Katague, Maria Fernanda Pacheco.

**Funding acquisition:** Gabriela Buccini.

**Methodology:** Gabriela Buccini.

**Supervision:** Gabriela Buccini, Rafael Pérez-Escamilla.

**Validation:** Gabriela Buccini.

**Writing – original draft:** Gabriela Buccini, Lily Kofke, Haley Case.

**Writing – review & editing:** Gabriela Buccini, Marina Katague, Maria Fernanda Pacheco, Rafael Pérez-Escamilla.

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
