## [Decision Letter · Decision Letter 0]

6 Feb 2023

PGPH-D-22-01250

Pathways to Scale-Up Early Childhood Programs: A Scoping Review of Reach Up and Care for Child Development

Dear Dr. Buccini,

Thank you for submitting your manuscript to PLOS Global Public Health. After careful consideration, we feel that it has merit but does not fully meet PLOS Global Public Health’s publication criteria as it currently stands. Therefore, we invite you to submit a revised version of the manuscript that addresses the points raised during the review process.

Please see comments from one reviewer below. Please note that we have only been able to secure a single reviewer to assess your manuscript. We are issuing a decision on your manuscript at this point to prevent further delays in the evaluation of your manuscript. Please be aware that the editor who handles your revised manuscript might find it necessary to invite additional reviewers to assess this work once the revised manuscript is submitted. However, we will aim to proceed on the basis of this single review if possible. 

We look forward to receiving your revised manuscript.

Kind regards,

Hanna Landenmark

Staff Editor

Journal Requirements:

b. If any authors received a salary from any of your funders, please state which authors and which funders.

3. Please ensure that Funding Information and Financial Disclosure Statement are matched.

4. We do not publish any copyright or trademark symbols that usually accompany proprietary names, eg  ©, ®, ™  (e.g. next to drug or reagent names). Please remove all instances of trademark/copyright symbols throughout the text, including ™ on page 7.

5. Fig 2: please (a) provide a direct link to the base layer of the map (i.e., the country or region border shape) and ensure this is also included in the figure legend; and (b) provide a link to the terms of use / license information for the base layer image or shapefile. We cannot publish proprietary or copyrighted maps (e.g. Google Maps, Mapquest) and the terms of use for your map base layer must be compatible with our CC-BY 4.0 license. 

Additional Editor Comments (if provided):

Reviewers' comments:

Reviewer's Responses to Questions

**Comments to the Author**

1. Does this manuscript meet PLOS Global Public Health’s publication criteria? Is the manuscript technically sound, and do the data support the conclusions? The manuscript must describe methodologically and ethically rigorous research with conclusions that are appropriately drawn based on the data presented.

Reviewer #1: Yes

2. Has the statistical analysis been performed appropriately and rigorously?

Reviewer #1: N/A

3. Have the authors made all data underlying the findings in their manuscript fully available (please refer to the Data Availability Statement at the start of the manuscript PDF file)?

Reviewer #1: No

4. Is the manuscript presented in an intelligible fashion and written in standard English?

Reviewer #1: Yes

5. Review Comments to the Author

Reviewer #1: This study provides a timely and critical summary of the implementation pathways of two widely-used nurturing care interventions. The authors use the findings of this review to provide a clear and necessary checklist to guide program implementers in their decision-making processes. I have included some comments in my review on minor points the authors can address to further improve the readability of their manuscript.

1. Please review the manuscript to correct minor spelling and grammar mistakes.

2. Results (pages 13-25) – When describing the implementation strategy themes, it would be helpful to have specific examples of some themes in the text (e.g., type of training skills used to train implementation agents) instead of putting all of those details in the tables. Right now, there is a lot of key information in tables, and it is difficult to go back and forth between the text and tables to clearly understand the results.

3. Results (pages 13-25) – Likewise, it would also be helpful if the authors provided more information on implementation outcomes including their definition (to ensure that readers and the authors have similar understandings of each term) and how they were assessed across studies.

4. The authors should consider including Luoto et al’s (2021) recent implementation evaluation paper (published in Frontiers in Public Health) in the review

5. Figure 10 – the details of the attached image are difficult to read, please provide a higher-quality image.

6. Discussion (page 36, lines 498-507) – The proposed ECD Program Implementation Checklist provides a helpful guide for decision-making on program implementation strategies. The authors do recommend the use of this checklist in conjunction with guidelines such as TIDieR and STaRI to reporting a program’s implementation details; however, these guidelines are limited with respect to reporting of implementation processes, intervention theory of change, fidelity, and program adaptation. Perhaps the authors should consider including the CARE guidelines (Yousafzai et al., 2018, Annals of the New York Academy of Sciences) which include this information and are more specific to nurturing care interventions for promoting early child development.

6. PLOS authors have the option to publish the peer review history of their article (what does this mean?). If published, this will include your full peer review and any attached files.

**Do you want your identity to be public for this peer review?** For information about this choice, including consent withdrawal, please see our Privacy Policy.

Reviewer #1: No

---

## [Decision Letter · Decision Letter 1]

25 Jun 2023

PGPH-D-22-01250R1

Pathways to Scale Up Early Childhood Programs: A Scoping Review of Reach Up and Care for Child Development

Dear Dr. Buccini,

Thank you for submitting your manuscript to PLOS Global Public Health. After careful consideration, we feel that it has merit but does not fully meet PLOS Global Public Health’s publication criteria as it currently stands. Therefore, we invite you to submit a revised version of the manuscript that addresses the points raised during the review process.

We look forward to receiving your revised manuscript.

Kind regards,

Nida Ziauddeen, PhD

Academic Editor

Journal Requirements:

Additional Editor Comments (if provided):

Reviewers' comments:

Reviewer's Responses to Questions

**Comments to the Author**

1. If the authors have adequately addressed your comments raised in a previous round of review and you feel that this manuscript is now acceptable for publication, you may indicate that here to bypass the “Comments to the Author” section, enter your conflict of interest statement in the “Confidential to Editor” section, and submit your "Accept" recommendation.

Reviewer #1: All comments have been addressed

Reviewer #2: (No Response)

2. Does this manuscript meet PLOS Global Public Health’s publication criteria? Is the manuscript technically sound, and do the data support the conclusions? The manuscript must describe methodologically and ethically rigorous research with conclusions that are appropriately drawn based on the data presented.

Reviewer #1: Yes

Reviewer #2: Yes

3. Has the statistical analysis been performed appropriately and rigorously?

Reviewer #1: N/A

Reviewer #2: Yes

4. Have the authors made all data underlying the findings in their manuscript fully available (please refer to the Data Availability Statement at the start of the manuscript PDF file)?

Reviewer #1: Yes

Reviewer #2: (No Response)

5. Is the manuscript presented in an intelligible fashion and written in standard English?

Reviewer #1: Yes

Reviewer #2: Yes

6. Review Comments to the Author

Reviewer #1: Thank you for addressing the comments raised and making the necessary revisions. This updated manuscript is improved and nearly ready for publication. During the revision process, another (systematic) review on the implementation and evaluation of CCD programs was published (Ahun et al., 2023, Frontiers in Public Health). I suggest that the authors read this new review and revise their manuscript accordingly to build on its findings and avoid unnecessary repetitions across publications.

Reviewer #2: This was quite a comprehensive and very useful summary of the Reach Up and Care for Child Development program implementation pathways. The main outcome appears to be an ECD implementation checklist with six building blocks, which flows well from the implementation pathways identified in the study. The authors also take care in noting the dearth of implementation details in the literature, and how this gap could affect findings. It seems as if many of the first reviewer’s comments were addressed already, so I have relatively few very minor comments. I think that the paper is fine to publish once the editor is satisfied that the reviewer comments were dealt with.

1. Re-read to correct minor spelling and grammar errors. These include but are not limited to:

a. Line 89: “resources constrained settings” should be “resource-constrained settings”

b. Line 90 should be: “… have been no previous attempts…”

c. Line 175: “A code structure ……, was” rather than “were”

d. Line 356: should be “…e.g. assess of skills…”

e. In Table 10 on page 31: IS2.5 “delivery sites” rather than “delivery site”. IS3.4 should be “only to” rather than “to only”

f. Line 473: “befit form” should be “benefit from”.

g. Line 491: scale-up should be hyphenated as it’s a noun.

h. Line 520: fille din should be filled in.

i. Line 563: remove the full stop after Acknowledgments.

j. Line 542: implementors should be implementers as in the rest of the text.

2. Figure 3 is still illegible. Perhaps redo it in landscape and spread it across 2 or 3 pages?

3. Figures 3 to 9 are text rather than figures. Why is this?

4. Counting up the n’s in lines 203 to 209 gives me 17 CCD and 13 RU but the text says there should be 18 and 15 respectively. The map in Figure 2 also appears to have 17 and 13. Rewrite the text so it is clear how the total n’s match up to the stated totals in line 203.

7. PLOS authors have the option to publish the peer review history of their article (what does this mean?). If published, this will include your full peer review and any attached files.

**Do you want your identity to be public for this peer review?** For information about this choice, including consent withdrawal, please see our Privacy Policy.

Reviewer #1: No

Reviewer #2: No

---

## [Editor Report · Decision Letter 2]

5 Jul 2023

Pathways to Scale Up Early Childhood Programs: A Scoping Review of Reach Up and Care for Child Development

PGPH-D-22-01250R2

Dear Dr. Buccini,

We are pleased to inform you that your manuscript 'Pathways to Scale Up Early Childhood Programs: A Scoping Review of Reach Up and Care for Child Development' has been provisionally accepted for publication in PLOS Global Public Health.

Best regards,

Nida Ziauddeen, PhD

Academic Editor